# Rapid whole brain imaging of neural activity in freely behaving larval zebrafish (*Danio rerio*)

Lin Cong[1†], Zeguan Wang[2†], Yuming Chai[2†], Wei Hang[1†], Chunfeng Shang[1], Wenbin Yang[2], Lu Bai[1,3], Jiulin Du[1,3], Kai Wang[1,3]*, Quan Wen[2]*

[1]Institute of Neuroscience, State Key Laboratory of Neuroscience, CAS Center for Excellence in Brain Science and Intelligence Technology, Shanghai Institutes for Biological Sciences, Chinese Academy of Sciences, Shanghai, China; [2]Center for Integrative Imaging, Hefei National Laboratory for Physical Sciences at Microscale, CAS Center for Excellence in Brain Science and Intelligence Technology, School of Life Sciences, University of Science and Technology of China, Hefei, China; [3]University of Chinese Academy of Sciences, Beijing, China

**Abstract** The internal brain dynamics that link sensation and action are arguably better studied during natural animal behaviors. Here, we report on a novel volume imaging and 3D tracking technique that monitors whole brain neural activity in freely swimming larval zebrafish (*Danio rerio*). We demonstrated the capability of our system through functional imaging of neural activity during visually evoked and prey capture behaviors in larval zebrafish.

DOI: https://doi.org/10.7554/eLife.28158.001

*For correspondence:
wangkai@ion.ac.cn (KW);
qwen@ustc.edu.cn (QW)

†These authors contributed equally to this work

Competing interests: The authors declare that no competing interests exist.

## Introduction

A central goal in systems neuroscience is to understand how distributed neural circuitry dynamics drive animal behaviors. The emerging field of optical neurophysiology allows monitoring (*Kerr and Denk, 2008*; *Dombeck et al., 2007*) and manipulating (*Wyart et al., 2009*; *Boyden et al., 2005*; *Zhang et al., 2007*) the activities of defined populations of neurons that express genetically encoded activity indicators (*Chen et al., 2013*; *Tian et al., 2009*) and light-activated proteins (*Kerr and Denk, 2008*; *Boyden et al., 2005*; *Zhang et al., 2007*; *Luo et al., 2008*). Larval zebrafish (*Danio rerio*) are an attractive model system to investigate the neural correlates of behaviors owing to their small brain size, optical transparency, and rich behavioral repertoire (*Friedrich et al., 2010*; *Ahrens and Engert, 2015*). Whole brain imaging of larval zebrafish using light sheet/two-photon microscopy holds considerable potential in creating a comprehensive functional map that links neuronal activities and behaviors (*Ahrens et al., 2012*; *Ahrens et al., 2013*; *Engert, 2014*).

Recording neural activity maps in larval zebrafish has been successfully integrated with the virtual reality paradigm: closed-loop fictive behaviors in immobilized fish can be monitored and controlled via visual feedback that varies according to the electrical output patterns of motor neurons (*Ahrens et al., 2012*; *Engert, 2012*). The behavioral repertoire, however, may be further expanded in freely swimming zebrafish whose behavioral states can be directly inferred and when sensory feedback loops are mostly intact and active. For example, it is likely that vestibular as well as proprioceptive feedbacks are perturbed in immobilized zebrafish (*Engert, 2012*; *Bianco et al., 2012*). The crowning moment during hunting behavior (*Bianco et al., 2011*; *Patterson et al., 2013*; *Trivedi and Bollmann, 2013*) — when a fish succeeds in catching a paramecium — cannot be easily replicated in a virtual reality setting. Therefore, whole brain imaging in a freely swimming zebrafish may allow optical interrogation of brain circuits underlying a range of less explored behaviors.

**eLife digest** How do neurons in the brain process information from the senses and drive complex behaviors? This question has fascinated neuroscientists for many years. It is currently not possible to record the electrical activities of all of the 100 billion neurons in a human brain. Yet, in the last decade, it has become possible to genetically engineer some neurons in animals to produce fluorescence reporters that change their brightness in response to brain activity and then monitor them under a microscope. In small animals such as zebrafish larvae, this method makes it possible to monitor the activities of all the neurons in the brain if the animal's head is held still. However, many behaviors – for example, catching prey – require movement, and no existing technique could image brain activity in enough detail if the animal's head was moving.

Cong, Wang, Chai, Hang et al. have now made progress towards this goal by developing a new technique to image neural activity across the whole brain of a zebrafish larva as it swims freely in a small water-filled chamber. The technique uses high-speed cameras and computer software to track the movements of the fish in three dimensions, and then automatically moves the chamber under the microscope such that the animal's brain is constantly kept in focus. The newly developed microscope can capture changes in neural activity across a large volume all at the same time. It is then further adapted to overcome problems caused by sudden or swift movements, which would normally result in motion blur. With this microscope set up, Cong et al. were able to capture, for the first time, activity from all the neurons in a zebrafish larva's brain as it pursued and caught its prey.

This technique provides a new window into how brain activity changes when animals are behaving naturally. In the future, this technique could help link the activities of neurons to different behaviors in several popular model organisms including fish, worms and fruit flies.

DOI: https://doi.org/10.7554/eLife.28158.002

Although whole brain functional imaging methods are available for head-fixed larval zebrafish, imaging a speeding brain imposes many technical challenges. Current studies on freely swimming zebrafish are either limited to non-imaging optical systems (*Naumann et al., 2010*) or wide field imaging at low resolution (*Muto et al., 2013*). While light sheet microscopy (LSM) has demonstrated entire brain coverage and single neuron resolution in restrained zebrafish (*Ahrens et al., 2013*), it lacks the speed to follow rapid fish movement. Moreover, in LSM, the sample is illuminated from its side, a configuration that is difficult to be integrated with a tracking system. Conventional light field microscopy (LFM) (*Broxton et al., 2013*; *Prevedel et al., 2014*) is a promising alternative due to its higher imaging speed; however, its spatial resolution is relatively low. Specialized LFMs for monitoring neural activity utilizing temporal information were also developed recently (*Pégard et al., 2016*; *Nöbauer et al., 2017*), which rely on spatiotemporal sparsity of fluorescent signals and cannot be applied to moving animals.

Here, we describe a fast 3D tracking technique and a novel volume imaging method that allows whole brain calcium imaging with high spatial and temporal resolution in freely behaving larval zebrafish. Zebrafish larvae possess extraordinary mobility. They can move at an instantaneous velocity up to 50 mm/s (*Severi et al., 2014*) and acceleration of 1 g (9.83 m/s$^2$). To continuously track fish motion, we developed a high-speed closed-loop system in which (1) customized machine vision software allowed rapid estimation of fish movement in both the *x-y* and *z* directions; and, (2) feedback control signals drove a high-speed motorized *x-y* stage (at 300 Hz) and a piezo *z* stage (at 100 Hz) to retain the entire fish head within the field of view of a high numerical aperture (25×, NA = 1.05) objective.

Larval zebrafish can make sudden and swift movements that easily cause motion blur and severely degrade imaging quality. To overcome this obstacle, we developed a new eXtended field of view LFM (XLFM). The XLFM can image sparse neural activity over the larval zebrafish brain at near single cell resolution and at a volume rate of 77 Hz, with the aid of genetically encoded calcium indicator GCamp6f. Furthermore, the implementation of flashed fluorescence excitation (200 μs in duration) allowed blur-free fluorescent images to be captured when a zebrafish moved at a speed up to 10 mm/s. The seamless integration of the tracking and imaging system made it possible to reveal rich whole brain neural dynamics during natural behavior with unprecedented resolution. We

demonstrated the ability of our system during visually evoked and prey capture behaviors in larval zebrafish.

## Results

The newly developed XLFM is based on the general principle of light field (*Adelson and Wang, 1992*) and can acquire 3D information from a single camera frame. XLFM greatly relaxed the constraint imposed by the tradeoff between spatial resolution and imaging volume coverage in conventional LFM. This achievement relies on optics and in computational reconstruction techniques. First, a customized lenslet array (*Figure 1a*, *Figure 1—figure supplement 1*) was placed at the rear pupil plane of the imaging objective, instead of at the imaging plane as in LFM. Therefore, in ideal conditions, a 2D spatially invariant point spread function (PSF) could be defined and measured; in practice, the PSF was approximately spatially invariant (see Materials and methods). Second, the aperture size of each micro-lens was decoupled from their interspacing and spatial arrangement, so that both the imaging volume and the resolution could be optimized simultaneously given the limited imaging sensor size. Third, multifocal imaging (*Abrahamsson et al., 2013*; *Perwass and Wietzke, 2012*) was introduced to further increase the depth of view by dividing the micro-lenses array into two groups whose focal planes were at different axial positions (*Figure 1b and c*, *Figure 1—figure supplements 3* and *4*). Fourth, a new computational algorithm based on optical wave theory was developed to reconstruct the entire 3D volume from one image (*Figure 1—figure supplement 5*) captured by a fast camera (see Materials and methods).

We first characterized the XLFM by imaging 0.5 µm diameter fluorescent beads. In our design, the system had ~ Ø800 µm in plane coverage (Ø is the diameter of the lateral field of view) and more than 400 µm depth of view, within which an optimal resolution of 3.4 µm × 3.4 µm × 5 µm could be achieved over a depth of 200 µm (*Figure 1—figure supplements 6* and *7*, Materials and methods). In the current implementation, however, the imaging performance suffered from the variation in the focal length of the micro-lenses (*Figure 1—figure supplement 8*), which led to spatial variance of the PSF. As a result, the reconstruction performance and the achievable optimal resolution were shown to degrade beyond the volume of Ø500 µm × 100 µm (*Figure 1—figure supplements 9* and *10*). To minimize the reconstruction time while assuring whole brain coverage (~250 µm thick), all imaging reconstructions were carried out over a volume of Ø800 µm × 400 µm.

We next characterized the imaging performance by considering more fluorescent light sources distributed within the imaging volume. The achievable optimal resolution depends on the sparseness of the sample, because the information captured by the image sensor was insufficient to assign independent values for all voxels in the entire reconstructed imaging volume. Given the total number of neurons (~80,000 [*Hill et al., 2003*]) in a larval zebrafish brain, we next introduced a sparseness index $\rho$, defined as the fraction of neurons in the brain active at a given instant, and used numerical simulation and our reconstruction algorithm to characterize the dependence of achievable resolution on $\rho$. We identified a critical $\rho_c \approx 0.11$, below which active neurons could be resolved at the optimal resolution (*Figure 1—figure supplement 11b*). As $\rho$ increased, closely clustered neurons could no longer be well resolved (*Figure 1—figure supplement 11c–d*). Therefore, sparse neural activity is a prerequisite in XLFM for resolving individual neurons at the optimal resolution. Moreover, the above characterization assumed an aberration and scattering free environment; complex optical properties of biological tissue could also degrade the resolution (*Ji, 2017*).

We demonstrated the capabilities of XLFM by imaging the whole brain neuronal activities of a larval zebrafish (5 d post-fertilization [dpf]) at a speed of 77 volumes/s and relatively low excitation laser exposure of 2.5 mW/mm$^2$ (*Figure 1d*, *Video 1*). The fluorescent intensity loss due to photobleaching reached ~50% when the zebrafish, which expressed pan-neuronal nucleus-labelled GCamp6f (huc: h2b-gcamp6f), was imaged continuously for ~100 min and over more than 300,000 volumes (*Figure 1—figure supplement 12*, *Video 2* and *3*). To test whether XLFM could monitor fast changes in neuronal dynamics across the whole brain at high resolution (close to single neuron level), we first presented the larval zebrafish, restrained in low melting point agarose, with visual stimulation (~2.6 s duration). We found that different groups of neurons in the forebrain, midbrain, and hindbrain were activated at different times (*Figure 1e–f*, *Video 1* and *4*), suggesting rapid sensorimotor transformation across different brain regions.

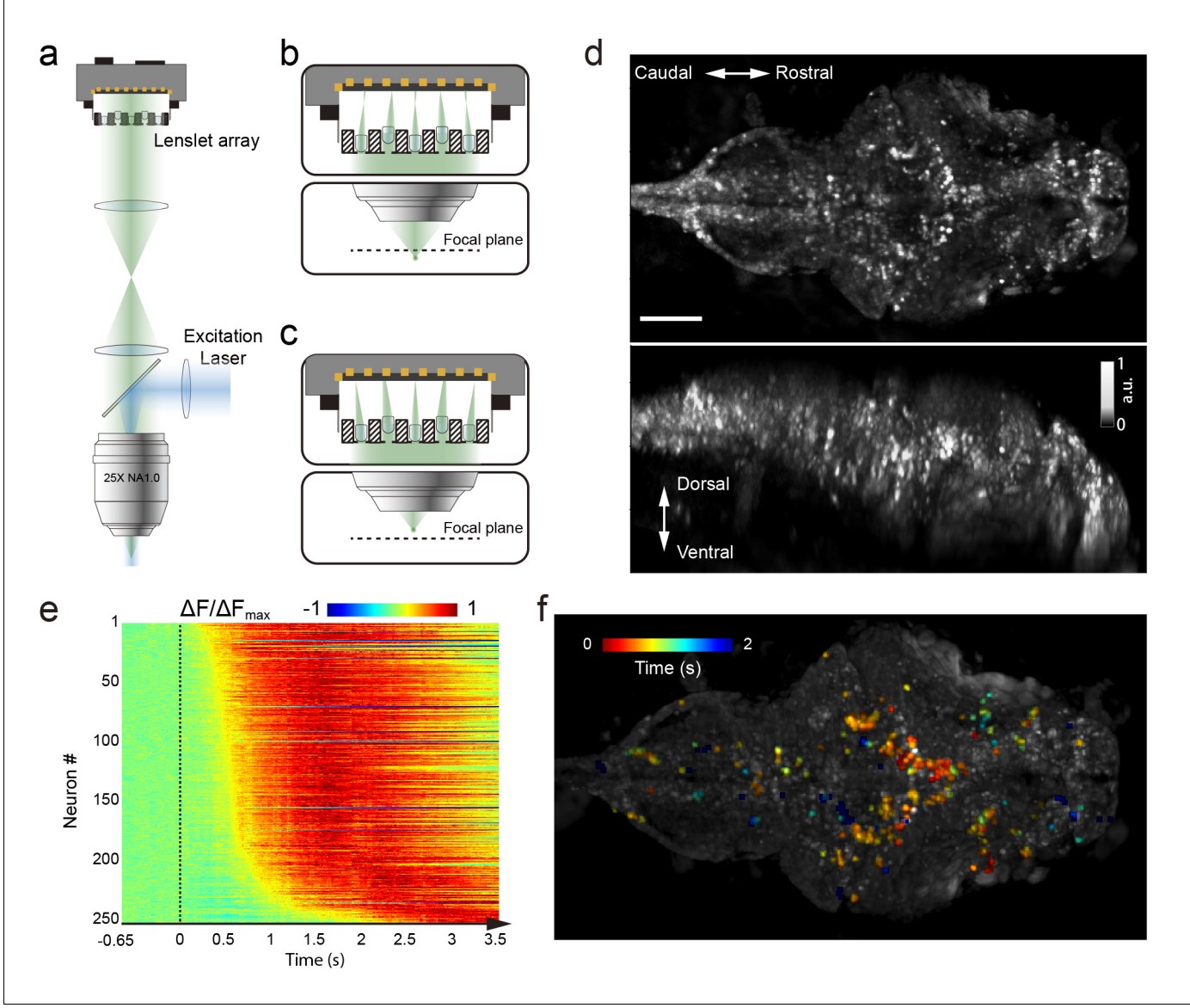

**Figure 1.** Whole brain imaging of larval zebrafish with XLFM. (**a**) Schematic of XLFM. Lenslet array position was conjugated to the rear pupil plane of the imaging objective. Excitation laser (blue) provided uniform illumination across the sample. (**b–c**) Point sources at two different depths formed, through two different groups of micro-lenses, sharp images on the imaging sensor, with positional information reconstructed from these distinct patterns. (**d**) Maximum intensity projections (MIPs) on time and space of time series volume images of an agarose-restrained larval zebrafish with pan-neuronal nucleus-localized GCaMP6f (huc:h2b-gcamp6f) fluorescence labeling. (**e**) Normalized neuronal activities of selected neurons exhibited increasing calcium responses after the onset of light stimulation at t = 0. Neurons were ordered by the onset time when the measured fluorescence signals reached 20% of their maximum. (**f**) Selected neurons in (**e**) were color coded based on their response onset time. Scale bar is 100 μm.
DOI: https://doi.org/10.7554/eLife.28158.003

The following figure supplements are available for figure 1:

**Figure supplement 1.** Customized lenslet array.
DOI: https://doi.org/10.7554/eLife.28158.004

**Figure supplement 2.** Experimentally measured PSF of the whole imaging system.
DOI: https://doi.org/10.7554/eLife.28158.005

**Figure supplement 3.** PSF of Group A micro-lenses: PSF_A.
DOI: https://doi.org/10.7554/eLife.28158.006

**Figure supplement 4.** PS F of Group B micro-lenses: PSF_B.
DOI: https://doi.org/10.7554/eLife.28158.007

**Figure supplement 5.** Example of camera captured raw imaging data of larval zebrafish.

*Figure 1 continued on next page*

*Figure 1 continued*

DOI: https://doi.org/10.7554/eLife.28158.008

**Figure supplement 6.** Characterization of in-plane resolution of micro-lenses.

DOI: https://doi.org/10.7554/eLife.28158.009

**Figure supplement 7.** Characterization of axial resolution of XLFM afforded by individual micro-lenses.

DOI: https://doi.org/10.7554/eLife.28158.010

**Figure supplement 8.** Characterization of magnification variation of micro-lenses in XLFM.

DOI: https://doi.org/10.7554/eLife.28158.011

**Figure supplement 9.** Resolution degradation due to focal length variation of micro-lenses.

DOI: https://doi.org/10.7554/eLife.28158.012

**Figure supplement 10.** Characterization of axial resolution of XLFM at low SNR.

DOI: https://doi.org/10.7554/eLife.28158.013

**Figure supplement 11.** Dependence of imaging resolution on the sparseness of the sample.

DOI: https://doi.org/10.7554/eLife.28158.014

**Figure supplement 12.** Characterization of photobleaching in fluorescence imaging by XLFM.

DOI: https://doi.org/10.7554/eLife.28158.015

To track freely swimming larval zebrafish, we transferred fish into a water-filled chamber with a glass ceiling and floor. The 20 mm × 20 mm × 0.8 mm-sized chamber was coupled with a piezo actuator and mounted on a high-speed 2D motorized stage (*Figure 2*). A tracking camera monitored the lateral movement of the fish, and an autofocus camera, which captured light field images, monitored the axial movement of the fish head (*Figure 2*, *Figure 2—figure supplement 1*).

Real-time machine vision algorithms allowed quick estimate of lateral (within 1 ms) and axial (~5 ms) head positions (see Materials and methods). The error signals in three dimensions, defined as the difference between the head position and set point, were calculated (*Figure 3a*) and converted to analog voltage signals through proportional-integral-derivative (PID) control to drive the motorized stage and z-piezo scanner. Tracking and autofocusing allowed for rapid compensation of 3D fish movement (300 Hz in x and y, 100 Hz in z, *Figure 3a*) and retainment of the fish head within the field of view of the imaging objective.

Our tracking system permitted high-speed and high-resolution recording of larval zebrafish behaviors. With two cameras acquiring head and whole body videos simultaneously (*Figure 2*, *Figure 3b*), we recorded and analyzed in real time (see Materials and methods) the kinematics of key features during larval zebrafish prey capture (*Figure 3b and c*, *Video 5* and *6*). Consistent with several earlier findings (*Bianco et al., 2011*; *Patterson et al., 2013*; *Trivedi and Bollmann, 2013*), eyes converged rapidly when the fish entered the prey capture state (*Figure 3c*). Other features that characterized tail and fin movement were also analyzed at high temporal resolution (*Figure 3c*).

The integration of the XLFM and 3D tracking system allowed us to perform whole brain functional imaging of a freely behaving larval zebrafish (*Figure 2*). We first replicated the light-evoked experiment (similar to *Figure 1*), albeit in a freely behaving zebrafish with pan-neuronal cytoplasm-labeled GCaMP6s (huc:gcamp6s), which exhibited faster and more prominent calcium response (*Video 7*). Strong activities were observed in the neuropil of the optical tectum and the midbrain after stimulus onset. The fish tried to avoid strong light exposure and made quick tail movement at ~60 Hz. Whole brain neural activity was monitored continuously during the light-evoked behavior, except for occasional

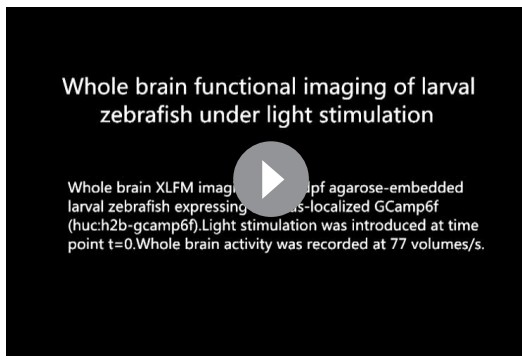

**Video 1.** Whole brain functional imaging of larval zebrafish under light stimulation. Whole brain XLFM imaging of a 5 dpf agarose-embedded larval zebrafish expressing nucleus-localized GCamp6f (huc:h2b-gcamp6f). Light stimulation was introduced at time point t = 0. Whole brain activity was recorded at 77 volumes/s.

DOI: https://doi.org/10.7554/eLife.28158.016

**Video 2.** Whole brain functional imaging of spontaneous activities of larval zebrafish. Whole brain XLFM imaging of a 5 dpf agarose-embedded larval zebrafish expressing nucleus-localized GCamp6f (huc: h2b-gcamp6f). Spontaneous neural activity was recorded at 0.6 volumes/s.
DOI: https://doi.org/10.7554/eLife.28158.017

**Video 3.** Whole brain functional imaging of spontaneous activities of larval zebrafish. Whole brain XLFM imaging of a 5 dpf agarose-embedded larval zebrafish expressing cytoplasm-labeled GCamp6s (huc: gcamp6s). Spontaneous neural activity was recorded at 0.6 volumes/s.
DOI: https://doi.org/10.7554/eLife.28158.018

**Video 4.** Whole brain functional imaging of larval zebrafish under light stimulation. Whole brain XLFM imaging of a 5 dpf agarose-embedded larval zebrafish expressing cytoplasm-labeled GCamp6s (huc: gcamp6s). Light stimulation was introduced at time point t = 0. Whole brain activity was recorded at 50 volumes/s.
DOI: https://doi.org/10.7554/eLife.28158.019

blurred frames due to the limited speed and acceleration of the tracking stage.

Next, we captured whole brain neural activity during the entire prey capture process in freely swimming larval zebrafish (huc:gcamp6s, *Video 8*). When a paramecium moved into the visual field of the fish, groups of neurons, indicated as group one in *Figure 4b*, near the contralateral optical tectum of the fish were first activated ($t_1$). The fish then converged its eyes onto the paramecium and changed its heading direction in approach ($t_2$). Starting from $t_2$, several groups of neurons in the hypothalamus, midbrain, and hindbrain, highlighted as groups two, three, and four in *Figure 4b*, were activated. It took the fish three attempts (*Figure 4c*) to catch and eat the paramecium. After the last try ($t_4$), neuron activity in group one decreased gradually, whereas activities in the other groups of neurons continued to rise and persisted for ~1 s before the calcium signals decreased. The earliest tectal activity (group 1) responsible for prey detection found here is consistent with previous studies (*Semmelhack et al., 2014*;

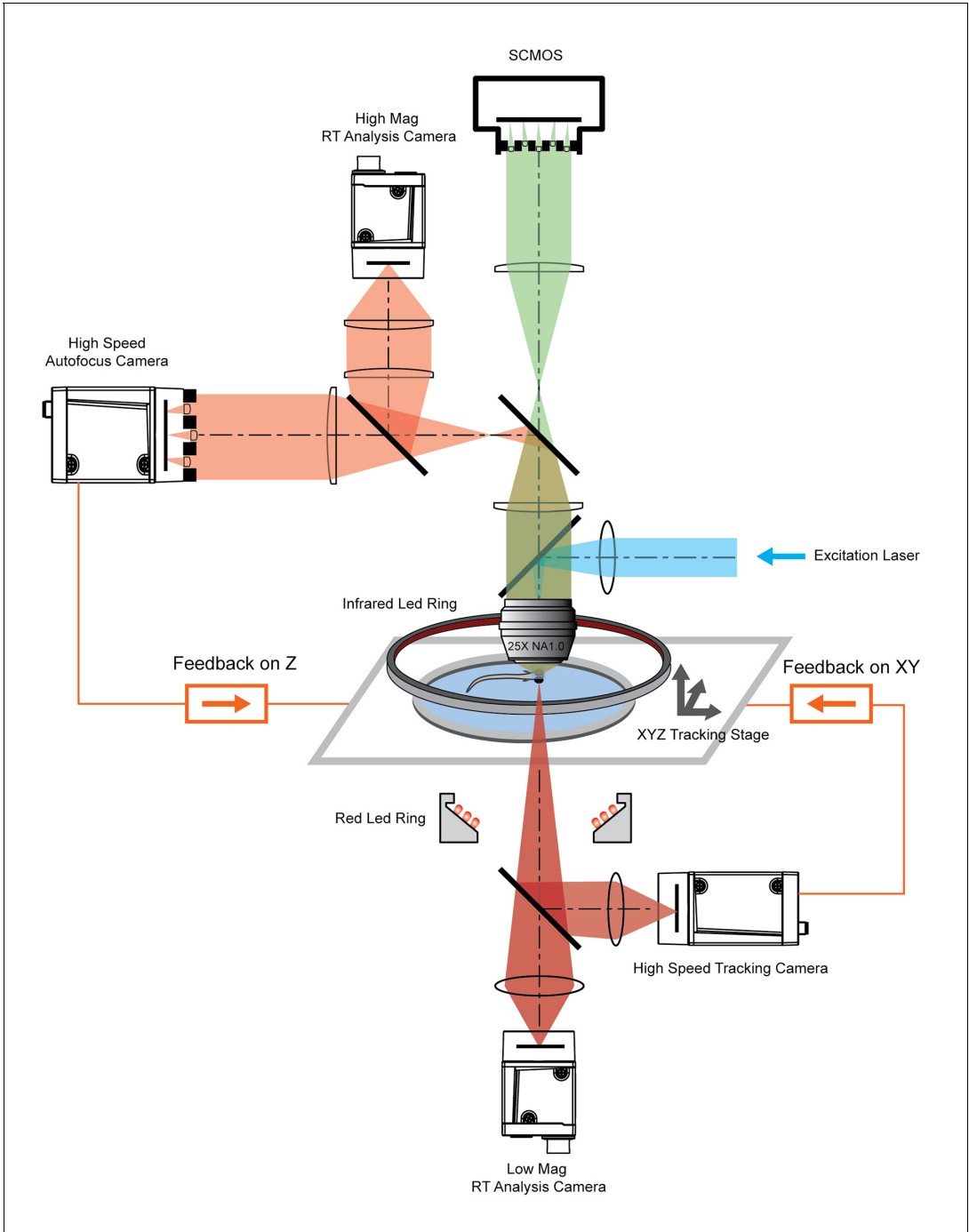

**Figure 2.** System schematics that integrated tracking, whole brain functional imaging, and real time behavioral analysis. Larval zebrafish swam in a customized chamber with an optically transparent ceiling and floor. The water-filled chamber was mounted on a high-speed three-axis stage (PI M686 and PI P725KHDS). Customized LED rings generated dark field illumination of the zebrafish. The scattered light was collected by four cameras: two cameras below the chamber were used for x-y plane tracking and low magnification real-time (RT) analysis, respectively; two cameras above the chamber and after the imaging objective were used for Z autofocus and high magnification RT analysis. The positional information of the larval zebrafish, acquired from the tracking and autofocus system, was converted to feedback voltage signals to drive the three-axis stage and to compensate for fish movement. The functional imaging system, described in *Figure 1*, shared the same imaging objective placed above the swimming chamber. The 3D tracking, RT behavioral analysis, and functional imaging system were synchronized for accurate correlation between neural activity and behavioral output.

DOI: https://doi.org/10.7554/eLife.28158.020

The following figure supplement is available for figure 2:

*Figure 2 continued on next page*

*Figure 2 continued*

**Figure supplement 1.** Characterization of the autofocus system.
DOI: https://doi.org/10.7554/eLife.28158.021

*Bianco and Engert, 2015*). Moreover, our data revealed interesting neural dynamics arising from other brain regions during and after successful prey capture. We also monitored similar behavior in a zebrafish expressing nucleus-localized GCamp6f (huc:h2b-gcamp6f) with better resolution but less prominent calcium response (*Video 9*).

## Discussion

Whole brain imaging in freely behaving animals has been previously reported in *Caenorhabditis elegans*, by integrating spinning-disk confocal microscopy with a 2D tracking system (*Venkatachalam et al., 2016*; *Nguyen et al., 2016*). In the more remote past, Howard Berg pioneered the use of 3D tracking microscopy to study bacteria chemotaxis (*Berg, 1971*). However, the significant increase of animal size imposes challenges both in tracking and imaging technologies. The XLFM, derived from the general concept of light field imaging (*Broxton et al., 2013*; *Adelson and Wang, 1992*; *Ng et al., 2005*; *Levoy et al., 2006*), overcomes several critical limitations of conventional LFM and allows optimization of imaging volume, resolution, and speed simultaneously. Furthermore, it can be perfectly combined with flashed fluorescence excitation to capture blur-free images at high resolution during rapid fish movement. Taken together, we have developed a volume imaging and tracking microscopy system suitable for observing and capturing freely behaving larval zebrafish, which have ~80,000 neurons and can move two orders of magnitude faster than *C. elegans*.

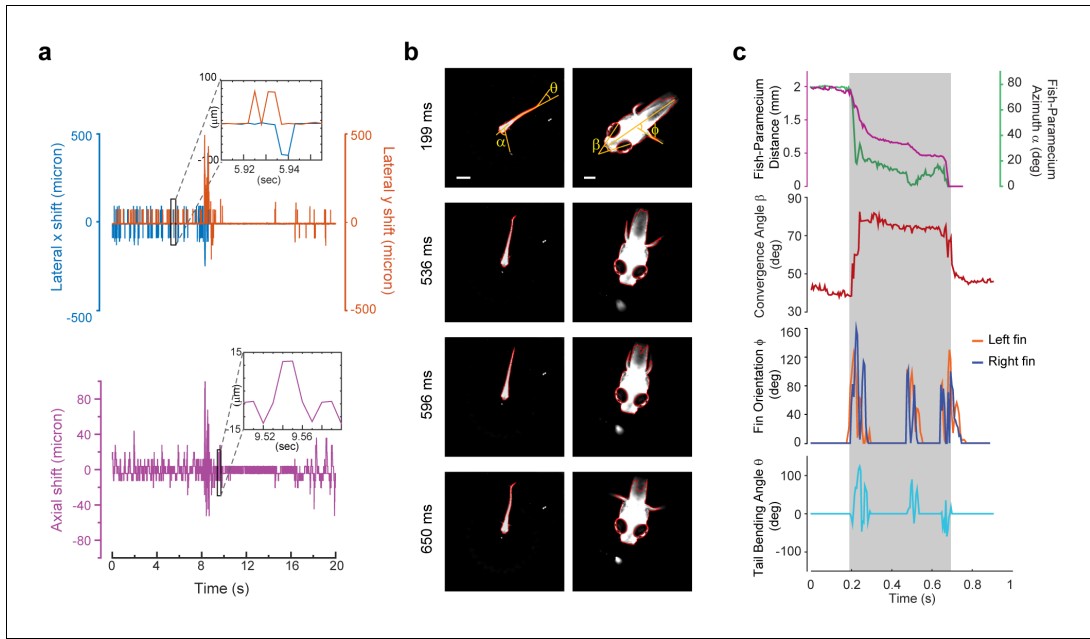

**Figure 3.** 3D tracking of larval zebrafish. (a) Representative time varying error signals in three dimensions, defined as the difference between real head position and set point. Inset provides magnified view at short time interval. Lateral movement can be rapidly compensated for within a few milliseconds with an instantaneous velocity of up to 10 mm/s. The axial shift was small compared with the depth coverage (200 μm) during whole brain imaging, and thereby had minor effect on brain activity reconstruction. (b) Tracking images at four time points during prey capture behavior, acquired at low (left) and high (right) magnification simultaneously. Scale bars are 1 mm (left) and 200 μm (right). (c) Kinematics of behavioral features during prey capture. Shaded region marks the beginning and end of the prey capture process.
DOI: https://doi.org/10.7554/eLife.28158.022

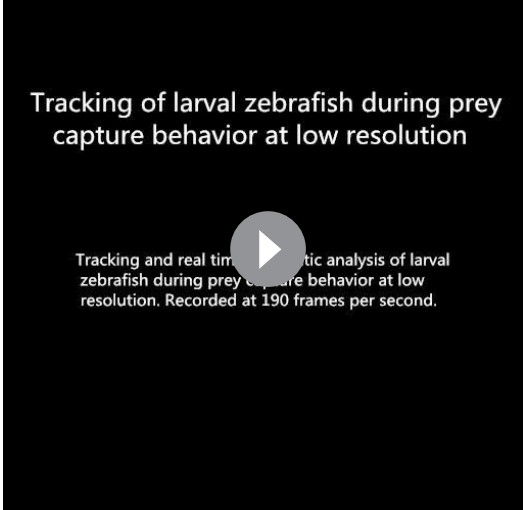

**Video 5.** Tracking of larval zebrafish during prey capture behavior at low resolution Tracking and real time kinematic analysis of larval zebrafish during prey capture behavior at low resolution. Recorded at 190 frames/s.
DOI: https://doi.org/10.7554/eLife.28158.023

**Video 6.** Tracking of larval zebrafish during prey capture behavior at high resolution. Tracking and real time kinematic analysis of larval zebrafish during prey capture behavior at high resolution. Recorded at 160 frames/s.
DOI: https://doi.org/10.7554/eLife.28158.024

Tracking and whole brain imaging of naturally behaving zebrafish provide an additional way to study sensorimotor transformation across the brain circuit. A large body of research suggests that sensory information processing depends strongly on the locomotor state of an animal (*Niell and Stryker, 2010*; *Maimon et al., 2010*; *Chiappe et al., 2010*). The ability to sense self-motion, such as proprioceptive feedback (*Pearson, 1995*) and efferent copy (*Bell, 1981*), can also profoundly shape the dynamics of the neural circuit and perception. To explore brain activity in swimming zebrafish, several studies have utilized an elegant tail-free embedding preparation (*Severi et al., 2014*; *Portugues and Engert, 2011*; *Portugues et al., 2014*), in which only the head of the fish is restrained in agarose for functional imaging. Nevertheless, it would be ideal to have physiological access to all neurons in defined behavioral states, where all sensory feedback loops remain intact and functional. Our XLFM-3D tracking system is one step towards this goal, and could be better exploited to explore the neural basis of more sophisticated natural behaviors, such as prey capture and social interaction, where the integration of multiple sensory feedbacks becomes critical.

In the XLFM, the camera sensor size limited the number of voxels and hence the number of neurons that could be reliably reconstructed. Our simulation suggested that the sparseness of neuronal activities is critical for optimal imaging volume reconstruction. A growing body of experimental data indeed suggests that population neuronal activities are sparse (*Hromádka et al., 2008*; *Buzsáki and Mizuseki, 2014*) and sparse representation is useful for efficient neural computation (*Olshausen and Field, 1996*; *Olshausen and Field, 2004*). Given the total number of neurons in the larval zebrafish brain, we found that when the fraction of active neurons in a given imaging frame was less than $\rho_c$

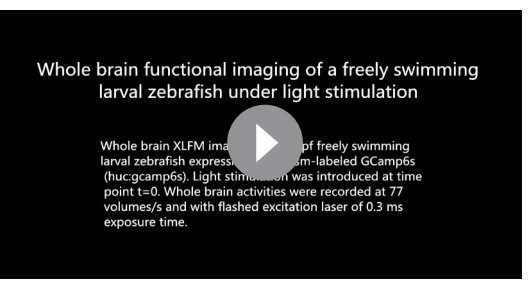

**Video 7.** Whole brain functional imaging of a freely swimming larval zebrafish under light stimulation Whole brain XLFM imaging of a 7 dpf freely swimming larval zebrafish expressing cytoplasm-labeled GCamp6s (huc:gcamp6s). Light stimulation was introduced at time point t = 0. Whole brain activities were recorded at 77 volumes/s and with a flashed excitation laser under 0.3 ms exposure time.
DOI: https://doi.org/10.7554/eLife.28158.025

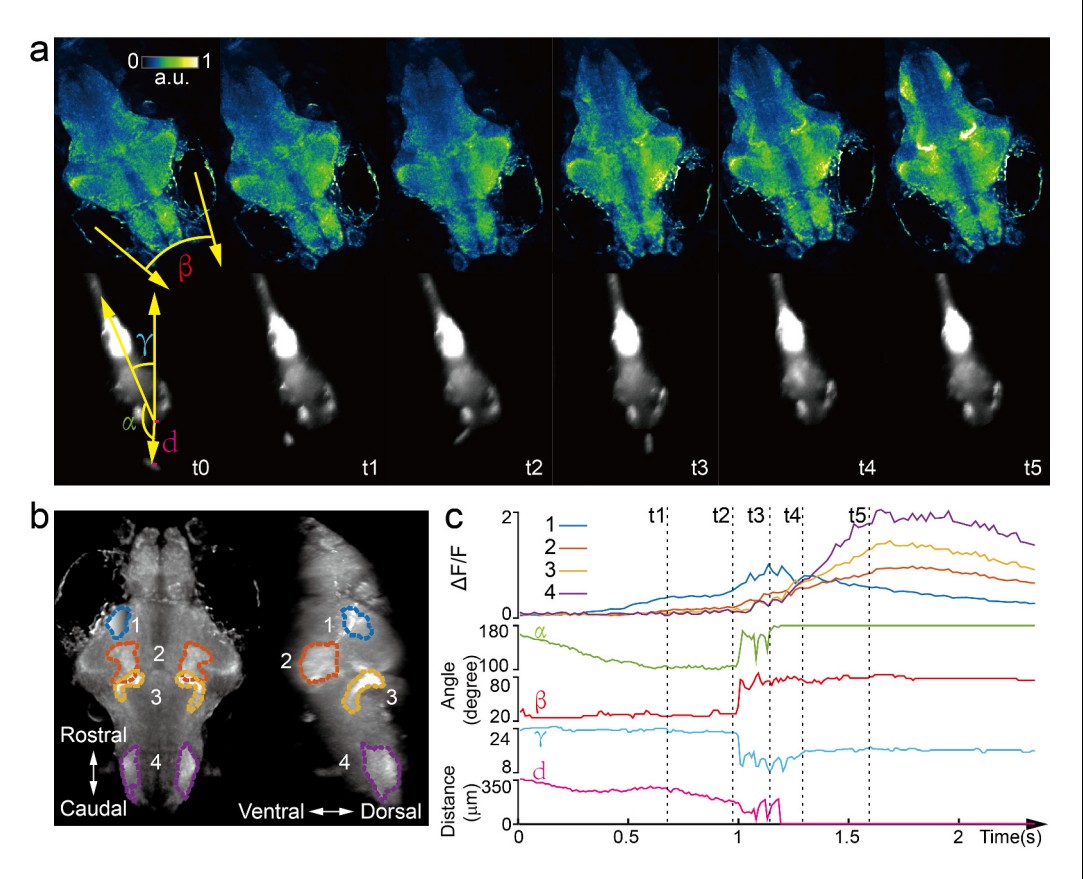

**Figure 4.** Whole brain imaging of larval zebrafish during prey capture behavior. (**a**) Renderings of whole brain calcium activity at six time points (up) and the corresponding behavioral images (bottom). Features used to quantify behavior were: fish-paramecium azimuth $\alpha$; convergence angle between eyes $\beta$; head orientation $\gamma$; and fish-paramecium distance $d$. (**b**) Maximum intensity projections of zebrafish brain with pan-neuronal cytoplasm-labeled GCaMP6s (huc:gcamp6s). Boundaries of four brain regions are color marked. (**c**) Neural dynamics inferred from GCaMP6 fluorescence changes in these four regions during the entire prey capture behavior (up) and the kinematics of behavioral features (bottom). Note that between t2 and t4, fish-paramecium distance $d$ exhibits three abrupt kinks, representing the three attempts to catch prey.

DOI: https://doi.org/10.7554/eLife.28158.026

$\approx 0.11$, individual neurons could be resolved at optimal resolution. When population neural activity was dense (*e.g.*, neurons have high firing rate and firing patterns have large spatiotemporal correlation), we obtained a coarse-grained neural activity map with reduced resolution.

To retain the fish head within the field of view of the imaging objective, our tracking system compensated for fish movement by continuously adjusting the lateral positions of the motorized stage. As a result, self-motion perceived by the fish was not exactly the same as that during natural behaviors. The linear acceleration of the swimming fish, encoded by vestibular feedback, was significantly underestimated. The perception of angular acceleration during head orientation remained largely intact. The relative flow velocity along the fish body, which was invariant upon stage translation, can still be detected by specific hair cells in the lateral line system (*Coombs, 2014*; *Liao, 2010*). Together, the interpretation of brain activity associated with self-motion must consider motion compensation driven by the tracking system.

Both tracking and imaging techniques can be improved in the future. For example, the current axial displacement employed by the piezo scanner had a limited travelling range (400 µm), and our swimming chamber essentially restrained the movement of the zebrafish in two dimensions. This limitation could be relaxed by employing axial translation with larger travelling range and faster

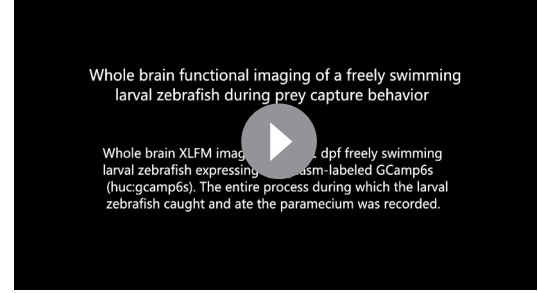

**Video 8.** Whole brain functional imaging of a freely swimming larval zebrafish during prey capture behavior. Whole brain XLFM imaging of an 11 dpf freely swimming larval zebrafish expressing cytoplasm-labeled GCamp6s (huc:gcamp6s). The entire process during which the larval zebrafish caught and ate the paramecium was recorded.
DOI: https://doi.org/10.7554/eLife.28158.027

**Video 9.** Whole brain functional imaging of a freely swimming larval zebrafish during prey capture behavior. Whole brain XLFM imaging of a 7 dpf freely swimming larval zebrafish expressing nucleus-localized GCamp6f (huc:h2b-gcamp6f). The entire process during which the larval zebrafish caught and ate the paramecium was recorded.
DOI: https://doi.org/10.7554/eLife.28158.028

dynamics. Furthermore, to avoid any potential disturbance of animal behaviors, it would be ideal if the imaging system moved, instead of the swimming chamber.

In XLFM, the performance degradation caused by focal length variation of the micro-lenses could be resolved by higher precision machining. In addition, the capability of XLFM could be further improved with the aid of technology development in other areas. With more pixels on the imaging sensor, we could resolve more densely labelled samples, and achieve higher spatial resolution without sacrificing imaging volume coverage by introducing more than two different focal planes formed by more groups of micro-lenses. With better imaging objectives that could provide higher numerical aperture and larger field of view at the same time, we could potentially image the entire nervous system of the larval zebrafish with single neuron resolution in all three dimensions. Additionally, the fast imaging speed of XLFM holds the potential for recording electrical activity when high signal-to-noise ratio (SNR) fluorescent voltage sensors become available (*St-Pierre et al., 2014*). Finally, the illumination-independent characteristic of XLFM is perfectly suitable for recording brain activities from bioluminescent calcium/voltage indicators in a truly natural environment, where light interference arising from fluorescence excitation can be eliminated (*Naumann et al., 2010*).

## Materials and methods

### XLFM

The imaging system (*Figure 1*) was a customized upright microscope. Along the fluorescence excitation light path, a blue laser (Coherent, OBIS 488 nm, 100 mW, USA) was expanded and collimated into a beam with a diameter of ~25 mm. It was then focused by an achromatic lens (focal length: 125 mm) and reflected by a dichroic mirror (Semrock, Di02-R488−25×36, USA) into the back pupil of the imaging objective (Olympus, XLPLN25XWMP2, 25X, NA 1.05, WD 2 mm, Japan) to result in an illumination area of ~1.44 mm in diameter near the objective's focal plane. In the fluorescence imaging light path, excited fluorescence was collected by the imaging objective and transmitted through the dichroic mirror. A pair of achromatic lenses (focal lengths: F1 = 180 mm and F2 = 160 mm), arranged in 2F1 +2F2, were placed after the objective and dichroic mirror to conjugate the objective's back pupil onto a customized lenslet array (*Figure 1—figure supplement 1*). The customized lenslet array was an aluminum plate with 27 holes (1.3 mm diameter aperture on one side and 1 mm diameter aperture on the other side, *Source code file 1*) housing 27 customized micro-lenses (1.3 mm diameter, focal length: 26 mm). The 27 micro-lenses were divided into two groups (*Figure 1—figure supplement 1*) and an axial displacement of 2.5 mm was introduced between them. Apertures of 1 mm diameter on the aluminum plate were placed right at the objective's pupil plane so that all micro-lenses samples light at pupil plane even though they were displaced axially after apertures. Due to the blockage of light by the aluminum micro-lenses housing, 16% of the light after a 1.05 NA

imaging objective was effectively collected by the camera. This efficiency is equivalent to using a 0.4 NA imaging objective. Finally, the imaging sensor of a sCMOS camera (Hamamatsu, Orca-Flash 4.0 v2, Japan) was placed at the middle plane between two focal planes formed by two different groups of micro-lenses. The total magnification of the imaging system was ~4, so one camera pixel (6.5 μm) corresponded to ~1.6 μm on the sample.

We developed a computational algorithm for 3D volume reconstruction, which required an accurately measured PSF (*Figure 1—figure supplement 2*). The PSF was measured by recording images of a 500 nm diameter fluorescent bead sitting on a motorized stage under the objective. A stack of 200 images was recorded when the bead was scanned with a step size of 2 μm in the axial direction from 200 μm below the objective's focal plane to 200 μm above. Since the images formed by two different groups of micro-lenses were from different axial locations and had different magnifications, the measured raw PSF data were reorganized into two complementary parts: PSF_A and PSF_B (*Figure 1—figure supplements 3* and *4*), according to the spatial arrangement of the micro-lenses. We took PSF_A stack, PSF_B stack, and a single frame of a raw image (2048 × 2048 pixels) as inputs, and applied a newly developed algorithm to reconstruct the 3D volume.

## Image reconstruction of XLFM

The reconstruction algorithm was derived from the Richardson-Lucy deconvolution. The goal was to reconstruct a 3D fluorescent object from a 2D image:

$$Obj(x,y,z)$$

The algorithm assumes that the real 3D object can be approximated by a discrete number of *x-y* planes at different *z* positions:

$$Obj(x,y,z) \sim Obj(x,y,z_k)$$

The numbers and positions of these planes can be arbitrary, yet the Nyquist sampling rate should be chosen to optimize the speed and accuracy of the reconstruction.

As the imaging system consisted of two different groups of micro-lenses (*Figure 1—figure supplement 1*), their PSFs (*Figure 1—figure supplements 3* and *4*) each consisted of a stack of planes that were measured at the same chosen axial positions $z_k$:

$$PSF_A(x,y,z_k)$$

Although the PSF was measured in imaging space, here we denote *x* and *y* as coordinates in object space to follow conventions in optical microscopy. Here and below, the combination of $PSF_A$ and $PSF_B$ is the total PSF.

Additionally, the images formed by two different groups of micro-lenses had different magnifications, which could be determined experimentally. The ratio between two different magnifications can be defined as:

$$\gamma = \frac{Magnification\ of\ group\ A\ microlenses}{Magnification\ of\ group\ B\ microlenses}$$

Then, the captured image on the camera can be estimated as:

$$Img_{Est}(x,y) = \sum_{k=1}^{n} \{Obj_A(x,y,z_k) \bigotimes PSF_A(x,y,z_k) + Obj_B(x,y,z_k) \bigotimes PSF_B(x,y,z_k)\},$$

where $Obj_A(x,y,z_k) = Obj_B(\gamma x, \gamma y, z_k)$

The operator $\bigotimes$ represents 2D convolution. Here, *x* and *y* on the left hand side of the equation also represent coordinates in object space so that 2D convolution was carried out in the same coordinates.

The goal of the algorithm is to estimate the $Obj(x,y,z_k)$ from the measured camera frame:

$$Img_{Meas}(x,y)$$

According to the Richardson-Lucy deconvolution algorithm, the iterative reconstruction can be expressed as:

$$Img_{Est}^{i}(x,y)=\sum_{k=1}^{n}\{Obj_A^{i-1}(x,y,z_k)\bigotimes PSF_A(x,y,z_k)+Obj_B^{i-1}(x,y,z_k)\bigotimes PSF_B(x,y,z_k)\}$$

$$Obj_A^{imp}(x,y,z_k)=Obj_A^{i-1}(x,y,z_k)\{\frac{Img_{Meas}(x,y)}{Img_{Est}^{i}(x,y)}\bigotimes PSF_A(-x,-y,z_k)\}$$

$$Obj_B^{imp}(x,y,z_k)=Obj_B^{i-1}(x,y,z_k)\{\frac{Img_{Meas}(x,y)}{Img_{Est}^{i}(x,y)}\bigotimes PSF_B(-x,-y,z_k)\}$$

$$Obj_A^{i}(x,y,z_k)=w(z_k)Obj_A^{imp}(x,y,z_k)+(1-w(z_k))Obj_B^{imp}(\gamma x,\gamma y,z_k)$$

$$Obj_B^{i}(x,y,z_k)=w(z_k)Obj_A^{imp}(\frac{x}{\gamma},\frac{y}{\gamma},z_k)+(1-w(z_k))Obj_B^{imp}(x,y,z_k)$$

Here, $0\leq w(z_k)\leq1$ is the weighting factor at different axial positions. The choice of $w(z_k)$ can be arbitrary. Because the resolutions achieved by different groups of micro-lenses at different z positions were not the same, the weighting factor can take this effect into consideration by weighing higher quality information more than lower quality information. One simple choice is $w(z_k)=0.5$, that is, to weigh information from two groups of micro-lenses equally.

The starting estimate of the object can be any non-zero value. Near the end of the iterations, $Obj_A^{i}(x,y,z_k)$ and $Obj_B^{i}(x,y,z_k)$ are interchangeable, except with different magnifications. Either can be used as the resulting estimate of the 3D object.

In XLFM, together with its reconstruction algorithm, the diffraction of the 3D light field is properly considered by experimentally measured PSF. The raw imaging data can be fed into the algorithm directly without any preprocessing. Given that the PSF is spatially invariant, which is satisfied apart from small aberrations, the algorithm can handle overlapping fish images (*Figure 1—figure supplement 5*). As a result, the field of view can be increased significantly. The reconstruction algorithm was typically terminated after 30 iterations when modifications in the estimated object became very small. The computation can speed up significantly via GPU. It took about 4 min to reconstruct one 3D volume using a desktop computer with a GPU (Nvidia Titan X). In comparison, the reconstruction ran ~20 × slower using a CPU (Intel E5-2630v2) on a Dell desktop. The source code written in MAT-LAB can be found in the *Source code file 2*.

The 3D deconvolution method has been developed for conventional LFM (*Broxton et al., 2013*). Our method differs from *Broxton et al. (2013)* in several ways. (1) The optical imaging systems are different. (2) The definitions of PSFs are different. Ours defines a spatially *invariant* PSF (see below for detailed characterization), whereas *Broxton et al. (2013)* defined a spatially variant PSF, leading to increased computational complexity in the deconvolution algorithm. (3) The PSF in *Broxton et al. (2013)* was simulated based on a model derived from an ideal imaging system, whereas ours was measured experimentally. Furthermore, our system took practical conditions, such as a non-ideal imaging objective, actual positions of microlenses, the spectrum of received fluorescence signal *et al.*, into consideration.

## Characterization of the spatial invariance of PSF in XLFM

The definition of a 2D spatially invariant PSF fundamentally means that in an ideal optical microscopy system, the resulting image can be described as a 2D convolution between object and PSF. As discussed in the previous section, this operation forms the basis of our reconstruction algorithm.

One of the fundamental differences between XLFM and conventional LFM is the location of the microlens array. In XLFM, the microlens array is placed at the pupil plane and the image sensor is at imaging plane, whereas in conventional LFM, the microlens array is placed at the image plane and the image sensor is| at pupil plane. It is possible to define a spatially invariant PSF in XLFM because:

1. Spatially invariant PSFs can be defined for individual sub-imaging systems consisting of different micro-lenses.
2. A spatially invariant PSF can be defined for the entire imaging system if the magnifications of all sub-imaging systems are the same.

By definition, the imaging formation in an ideal optical imaging system is linear and spatially invariant, so spatially invariant PSFs for sub-imaging systems consisting of micro-lens *A1* and *A2* can be defined as:

$$Image_{A1} = Object \bigotimes PSF_{A1}$$

$$Image_{A2} = Object \bigotimes PSF_{A2}$$

where $Image_{A1/2}$ are sub-images behind individual micro-lens. If we perform the convolution in the imaging space, the coordinates of $Object(x, y)$ should be scaled by the magnification factors of their sub-image systems, respectively. Now if the magnifications of different sub-image systems are the same, the summation of all PSFs formed by individual micro-lenses can be defined as a single PSF. In other words,

$$Image_A = Image_{A1} + Image_{A2} = Object \bigotimes (PSF_{A1} + PSF_{A2}) = Object \bigotimes PSF_A$$

$$\text{where } PSF_A = PSF_{A1} + PSF_{A2}$$

Experimentally, the small variation of individual micro-lenses' focal length (*Figure 1—figure supplement 8*) resulted in spatial variance of $PSF_A$ or $PSF_B$, but it does not affect the imaging formation theory of XLFM. The spatial variance led to degraded reconstruction performance, as shown in *Figure 1—figure supplement 9*. This degradation was negligible near the center of the field of view, but became more evident near the edge of the field of view. This is because the PSF was measured near the center of the field of view. The reconstruction algorithm produces 27 estimates of the same object based on 27 sub-images. In the meanwhile, it tries to combine and align these estimates all together in the same coordinates. The position where the PSF is measured determines the origin of this coordinates. If the magnifications of different micro-lenses are different, the reconstruction will yield an image that is clear near the origin of the coordinates but blurred at the edge, as shown in in *Figure 1—figure supplement 9*.

## Resolution characterization of XLFM

Unlike conventional microscopy, where the performance of the imaging system is fully characterized by the PSF at the focal plane, the capability of XLFM is better characterized as a function of positions throughout the imaging volume.

We first characterized the spatial resolution in the *x-y* plane by analyzing the spatial frequency support of the experimentally measured PSF from individual micro-lenses using a 0.5 µm diameter fluorescent bead. The optical transfer function (OTF), which is the Fourier transform of the PSF in the *x-y* plane, was extended to a spatial frequency of ~1/3.4 µm$^{-1}$ (*Figure 1—figure supplement 6*), a result that agreed well with the designed resolution at 3.4 µm, given that the equivalent NA of individual micro-lenses was 0.075.

The lateral resolution, measured from the raw PSF behind individual micro-lenses, was preserved across the designed cylindrical imaging volume of Ø800 µm × 200 µm (*Figure 1—figure supplement 6*). However, the reconstruction results (*Figure 1—figure supplement 9*), which used total PSF (*Figure 1—figure supplement 2*), exhibited resolution degradation when the fluorescent bead was placed more than 250 µm away from the center (*Figure 1—figure supplement 9*). This discrepancy resulted from the variation in focal length of the micro-lenses (*Figure 1—figure supplement 8*), which, in turn, led to spatial variance of the defined $PSF_A$ and $PSF_B$. In principle, the designed lateral resolution of 3.4 µm could be preserved over a volume of Ø800 µm × 200 µm by reducing focal length variation to below 0.3%

We next characterized the axial resolution of the XLFM. The XLFM gained axial resolution by viewing the object from large projection angles achieved by micro-lenses sitting near the edge of the objective's back pupil plane. For example, if two points of light source were located at the same position in the *x-y* plane, but were separated by *z* in the axial direction, then one micro-lens in the XLFM could capture an image of these two points with a shift between them. The shift can be determined as:

$$d = z * tan\theta$$

where $\theta$ is the inclination angle inferred from the measured PSF (*Figure 1—figure supplement 2*). If the two points in the image can be resolved, the two points separated by $z$ can be resolved by the imaging system. Since a micro-lens sitting in the outer layer of the array offered the largest inclination angle of 40 degree in our system, the axial resolution $dz$ can be directly calculated as:

$$dz = \frac{dxy}{tan\theta_{max}} = \frac{3.4\ \mu m}{tan(40°)} = 4\ \mu m$$

The best way to confirm the theoretical estimate is to image two fluorescent beads with precisely controlled axial separations. However, this is technically very challenging. Instead, we pursued an alternative method that is equivalent to imaging two beads simultaneously:

1. We took a z stack of images of fluorescent beads, as done in measuring the PSF.
2. In post processing, we added two images from different z positions to mimic the beads being present simultaneously at two different *z* positions.

The above method allowed us to experimentally characterize the axial resolution afforded by individual micro-lenses focusing at different z positions. We used a single fluorescent bead (0.5 $\mu m$ in diameter) with a high SNR (*Figure 1—figure supplement 7a*). We imaged at different axial positions: $z = -100$ $\mu m$, $z = 0$ $\mu m$, and $z = 100$ $\mu m$ (*Figure 1—figure supplement 7b*). The third column is the combined images in column 1 and 2. The capability of resolving the two beads in the third column can be demonstrated by spatial frequency analysis (fourth column in *Figure 1—figure supplement 7b*). The two line dips, indicating the existence of two beads instead of one rod in the fourth column, were confirmations of the resolving capability. This becomes more evident after deconvolution of the raw images (fifth column in *Figure 1—figure supplement 7b*). Micro-lenses 1 and 2 could resolve two beads, separated by 5 $\mu m$, within the range of $-100$ $\mu m \leq z \leq 0$ and $0 \leq z \leq 100$ $\mu m$, respectively. In other words, the complementary information provided by the two micro-lenses allowed the system to maintain a high axial resolution at 5 $\mu m$ across a 200 $\mu m$ depth.

Next, we imaged densely packed fluorescent beads (0.5 $\mu m$ in diameter) with a low SNR (*Figure 1—figure supplement 10a*), and used our reconstruction algorithm to determine the minimum axial separation between beads that could be resolved (*Figure 1—figure supplement 10b–c*). In this case, 5 $\mu m$ axial resolution could be preserved across a depth of 100 $\mu m$. The resolution decayed gradually to ~10 $\mu m$ at the edge of an imaging volume with a 400 $\mu m$ axial coverage (*Figure 1—figure supplement 10b*). We believe that the optimal axial resolution at 5 µm could be achieved over an axial coverage of 200 µm by minimizing micro-lens focal length variation (*Figure 1—figure supplement 8*).

Finally, we characterized how the imaging performance depended upon the sparseness of the sample. Given the total number of neurons (~80,000) in a larval zebrafish brain, we introduced a sparseness index $\rho$, defined as the fraction of neurons in the brain active at an imaging frame, and used numerical simulation to characterize the dependence of achievable resolution on $\rho$. To this end, we simulated a zebrafish larva with uniformly distributed firing neurons (red dots in *Figure 1—figure supplement 11a*). By convolving the simulated zebrafish with the experimentally measured PSFs (*Figure 1—figure supplements 3* and *4*), we generated an image that mimicked the raw data captured by the camera. We then reconstructed the simulated neurons from this image, represented by green dots. When $\rho$ was equal to or less than 0.11, which corresponded to ~9000 neurons activated at a given instant, all active neurons, including those closely clustered, could be reconstructed with optimal resolution (*Figure 1—figure supplement 11b* inset). As the sparseness index $\rho$ increased, the resolution degraded: nearby neurons merged laterally and elongated axially (*Figure 1—figure supplement 11c–d*). In all calculations, the Poisson noise was properly considered by assuming that each active neuron emitted 20,000 photons, 2.2% of which were collected by our imaging system.

In vivo resolution characterization is challenging due to a lack of bright and spot-like features in living animals. Additionally, achievable resolution depends on the optical properties of biological tissues, which can be highly heterogeneous and difficult to infer. The light scattering and aberration

induced by biological tissue usually leads to degraded imaging performance (*Ji, 2017*; *Ji et al., 2010*; *Wang et al., 2014*; *Wang et al., 2015*).

## XY tracking system

To compensate for lateral fish movement and retain the entire fish head within the field of view of a high NA objective (25×, NA = 1.05), a high-speed camera was used to capture fish motion (2 ms exposure time, 300 fps or higher, Basler aca2000-340kmNIR, Germany). We developed an FPGA-based RT system in LabVIEW that could rapidly identify the head position by processing the pixel stream data within the Cameralink card before the whole image was transferred to RAM. The error signal between the actual head position and the set point was then fed into the PID to generate output signals and control the movement of a high-speed motorized stage (PI M687 ultrasonic linear motor stage, Germany). In the case of large background noise, we alternatively performed conventional imaging processing in C/C++ (within 1 ms delay). The rate-limiting factor of our lateral tracking system was the response time of the stage (~300 Hz).

## Autofocus system

We applied the principle of LFM to determine the axial movement of larval zebrafish. The autofocus camera (100 fps or higher, Basler aca2000-340kmNIR, Germany) behind a one-dimensional microlens array captured triplet images of the fish from different perspectives (*Figure 2—figure supplement 1a*). Z motion caused an extension or contraction between the centroids of the fish head in the left and right sub-images, an inter-fish distance (*Figure 2—figure supplement 1b*) that can be accurately computed from image autocorrelation. The inter-fish distance, multiplied by a pre-factor, can be used to estimate the z position of the fish, as it varies linearly with axial movement (*Figure 2—figure supplement 1c*). The error signal between the actual axial position of the fish head and the set point was then fed into the PID to generate an output signal to drive a piezo-coupled fish container. The feedback control system was written in LabVIEW. The code was further accelerated by parallel processing and the closed loop delay was ~5 ms. The rate-limiting factor of the autofocus system was the settling time of the piezo scanner (PI P725KHDS, Germany, 400 μm travelling distance), which was about 10 ms.

## Real-time behavioral analysis

Two high-speed cameras acquired dark-field images at high and low magnification, respectively, and customized machine vision software written in C/C ++ with the aid of OpenCV library was used to perform real-time behavioral analysis of freely swimming larval zebrafish. At high magnification, eye positions, their orientation, and convergence angle were computed; at low magnification, the contour of the whole fish, centerline, body curvature, and bending angle of the tail were computed. The high mag RT analysis was run at ~120 fps and the low mag RT analysis was run at ~180 fps. The source code can be found in the *Source code file 3*.

## Ethics statement and animal handling

All animal handling and care were conducted in strict accordance with the guidelines and regulations set forth by the Institute of Neuroscience, Chinese Academy of Sciences, University of Science and Technology of China (USTC) Animal Resources Center, and University Animal Care and Use Committee. The protocol was approved by the Committee on the Ethics of Animal Experiments of the USTC (permit number: USTCACUC1103013).

All larval zebrafish (huc:h2b-gcamp6f and huc:gcamp6s) were raised in embryo medium under 28.5°C and a 14/10 hr light/dark cycle. Zebrafish were fed with paramecium from 4 dpf. For restrained experiments, 4–6 dpf zebrafish were embedded in 1% low melting point agarose. For freely moving experiments, 7–11 dpf zebrafish with 10% Hank's solution were transferred to a customized chamber (20 mm in diameter, 0.8 mm in depth), and 10–20 paramecia were added before the chamber was covered by a coverslip.

## Neural activity analysis

To extract neural activity induced by visual stimuli (*Figure 1e and f*), time series 3D volume stacks were first converted to a single 3D volume stack, in which each voxel represented variance of voxel

values over time. Candidate neurons were next extracted by identifying local maxima in the converted 3D volume stack. The region-of-interest (ROI) was set according to the empirical size of a neuron. The voxels around the local maxima were selected to represent neurons. The fluorescence intensity over each neuron's ROI was integrated and extracted as neural activity. Relative fluorescent changes $F/F_0$ were normalized to their maximum calcium response $F_{max}/F_0$ over time, and sorted according to their onset time when $F$ first reached 20% of its $F_{max}$ (*Figure 1e and f*) after the visual stimulus was presented.

## Visual stimulation

A short wavelength LED was optically filtered (short-pass optical filter with cut-off wavelength at 450 nm, Edmund #84–704) to avoid light interference with fluorescence. It was then focused by a lens into a spot 2 ~ 3 mm in diameter. The zebrafish was illuminated from its side. The total power of the beam was roughly 3 mW.

## Statement of replicates and repeats in experiments

Each experiment was repeated at least three times with similar experimental conditions. Imaging and video data acquired from behaviorally active larval zebrafish with normal huc:h2b-gcamp6f or huc:gcamp6s expression were used in the main figures and videos.

## Acknowledgements

We thank Misha B Ahrens for the zebrafish lines. We thank Yong Jiang, Tongzhou Zhao, WenKai Han, Shenqi Fan for assistance in building the 3D tracking system, real time behavioral analysis, and larval zebrafish experiments. We thank Dr Bing Hu and Dr Jie He for his support in zebrafish handling and helpful discussions.

## Additional information

### Funding

| Funder | Grant reference number | Author |
|---|---|---|
| Strategic Priority Research Program of the Chinese Academy of Sciences | XDB02060012 | Kai Wang |
| National Science Foundation of China | NSFC-31471051 | Quan Wen |
| China Thousand Talents Program | | Kai Wang |
| CAS Pioneer Hundred Talents Program | | Quan Wen |

The funders had no role in study design, data collection and interpretation, or the decision to submit the work for publication.

### Author contributions

Lin Cong, Wei Hang, Designed and built the XLFM, Designed and built the autofocus system, Did experiments under the supervision of Chunfeng Shang, Jiulin Du, Kai Wang and Quan Wen, Worked collaboratively to integrate the XLFM and the tracking system; Zeguan Wang, Designed and built the XLFM, Designed and built the X-Y tracking and the real-time behavioral analysis system, Designed and built the autofocus system, Did experiments under the supervision of Chunfeng Shang, Jiulin Du, Kai Wang and Quan Wen, Worked collaboratively to integrate the XLFM and the tracking system; Yuming Chai, Designed and built the X-Y tracking and the real-time behavioral analysis system, Did experiments under the supervision of Chunfeng Shang, Jiulin Du, Kai Wang and Quan Wen, Worked collaboratively to integrate the XLFM and the tracking system; Chunfeng Shang, Jiulin Du, Designed zebrafish behavioral experiments, Worked collaboratively to integrate the XLFM and the tracking system; Wenbin Yang, Designed and built the X-Y tracking and the real-time

behavioral analysis system, Worked collaboratively to integrate the XLFM and the tracking system; Lu Bai, Designed and built the XLFM, Worked collaboratively to integrate the XLFM and the tracking system; Kai Wang, Conceived the project, Conceived the idea of XLFM, Designed and built the XLFM, Designed and built the autofocus system, Wrote the paper with inputs from all authors, Worked collaboratively to integrate the XLFM and the tracking system; Quan Wen, Conceived the project, Designed and built the X-Y tracking and the real-time behavioral analysis system, Designed zebrafish behavioral experiments, Wrote the paper with inputs from all authors, Worked collaboratively to integrate the XLFM and the tracking system

### Author ORCIDs
Kai Wang (iD) https://orcid.org/0000-0001-7858-944X
Quan Wen (iD) https://orcid.org/0000-0003-0268-8403

### Ethics
Animal experimentation: Zebrafish handling procedures were approved by the Institute of Neuroscience, Shanghai Institutes for Biological Sciences, Chinese Academy of Sciences.(permit number: USTCACUC1103013).

### Decision letter and Author response
Decision letter https://doi.org/10.7554/eLife.28158.034
Author response https://doi.org/10.7554/eLife.28158.035

## Additional files

### Supplementary files
• Source code 1. Computer-Aided design files of mounting plates for micro-lenses array.
DOI: https://doi.org/10.7554/eLife.28158.029

• Source code 2. Source code for XLFM reconstruction.
DOI: https://doi.org/10.7554/eLife.28158.030

• Source code 3. Source code for Real-Time behavioral analysis.
DOI: https://doi.org/10.7554/eLife.28158.031

• Supplement file 1 Acquisition parameters for fluorescence imaging.
DOI: https://doi.org/10.7554/eLife.28158.032

• Transparent reporting form
DOI: https://doi.org/10.7554/eLife.28158.033

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
