## [Decision Letter]

Thank you for submitting your article "Rapid whole brain imaging of neural activities in freely behaving larval zebrafish" for consideration by *eLife*. Your article has been reviewed by three peer reviewers, one of whom, Ronald L Calabrese (Reviewer #1), is a member of our Board of Reviewing Editors, and the evaluation has been overseen by Eve Marder as the Senior Editor.

The reviewers have discussed the reviews with one another and the Reviewing Editor has drafted this decision to help you prepare a revised submission.

Summary:

This is an exciting manuscript, which reports a new development in Light Field Microscopy (LFM). The authors developed a Light Field Microscope: eXtended (XLFM) field of view seamlessly integrated with an X-Y tracking system and an auto focus. XLFM can simultaneously image intact brain neural activities (over a volume of 800 µm X 800 µm X 200 µm) at ~ 3.4 µm X 3.4 µm X 5 µm spatial resolution and at 77 Hz volume rate, with the aid of genetically encoded calcium indicator GCamp6f in a freely moving larval zebrafish during visual stimulation and prey capture. They provide stunning videos and enough processed data to show the value of the new development for imaging activity across the brain during real behavior.

The work is nicely illustrated with exemplar data. This is not a full report on the science behind the experiments illustrated but rather a proof of principle. Exciting science is in the offing but a new technology is showcased, as is appropriate for a Tools and Resources paper.

Essential revisions:

1) We find this technology to be a significant advance. There are several technical issues, however, that must be resolved. Further clarifications to the text are needed about precisely what was done and how it was done. Some claims need to be more carefully worded to recognize the limitations of the technique and recognize other contributions. The writing should be improved. The expert reviews provide a detailed list of all the points that should be considered in revision. Rather than paraphrasing those reports, they are included in full to ensure that the detailed technological issues are well-stated.

2) As stated explicitly in the expert reviews, software and design features must be made fully available to the scientific community with publication.

*Reviewer #1:*

1) The chamber in which the freely moving larva swims is ONLY 0.8 mm deep. Thus the animal is sandwiched between glass plates with no real ability to move in the z-direction. Essentially, it moves in two dimensions. The authors should address this limitation in their approach.

*Reviewer #2:*

In Cong et al., two advances are reported. First, a tracking system is introduced capable of keeping a freely swimming larval zebrafish in one location most of the time. Second, a new form of light field microscopy is reported capable of fast 3D imaging. Putting these together constitutes a system for whole-brain imaging in freely swimming zebrafish larvae, with a resolution slightly below single-cell.

In my opinion this is a major advance and I am supportive of publication in *eLife* with a few improvements.

For background, previous efforts to perform whole-brain imaging in behaving animals consisted of light-sheet (slower than light-field) in head-restrained animals, or light-field (a variant with, I believe, lower spatial resolution) in head-restrained animals. Imaging in freely behaving animals has been done in *C. elegans*, which move more slowly than zebrafish. Thus, compared to previous work, the advances of this manuscript are considerable. Furthermore, imaging most of the brain in freely swimming animals in really impressive.

Points that should be addressed:

1) The authors claim that the point spread function (PSF) is spatially invariant. This appears to be true if one considers the microscope an ideal optical system, but with non-ideal optics, it's unlikely. Even with a good objective, the entire system contributes to the PSF, and it's unlikely that all microlenses are diffraction limited over the entire field of view. Moreover, differential distortion between the sub-images would cause the full camera PSF to warp as the point source moves in the sample. So if Richardson-Lucy deconvolution only works with a spatially invariant PSF, and the true PSF is not fully spatially invariant, the question arises, What image artifacts do you get?

There may be multiple ways to answer this question. One path might be to (a) move a bead around a few x-y locations, including the extreme ones, and check how spatially invariant the PSF really is (and include the raw PSF volumes in the manuscript, e.g. by measuring them at two extreme points, shifting one by the predicted amount, and overlaying the two in different colors). (b) Next, assuming a spatially invariant PSF derived from one of the bead locations (e.g. the center), reconstruct a bead positioned at various points, including the edges of the volume, and quantify the spread of the point source in the reconstructed volume (this should have high brightness at the original bead position, plus dimmer pixel values spread at other locations, which should be quantified).

2) A follow-up: if the PSF is not fully spatially invariant, what does this mean for the statement that overlapping sub-images are permitted (subsection “Image reconstruction of XLFM”)? My understanding is that the overlap is fine so long as the PSF is fully x-y invariant, and if not, then some artifacts will be introduced. The reasons and assumptions underlying this statement should be clarified in the text.

For clarity, points (25) and (17) are not criticisms of the system, only a call for characterization of the artifacts that the reconstruction algorithm introduces when using simplifying assumptions.

3) The reconstruction algorithm (subsection “Image reconstruction of XLFM”) contains confusing notation (if I understand it correctly). The coordinates (x,y) on the left hand side of Equation 5, refer to image coordinates. But (x,y,z_k) on the right hand side refers to coordinates in the 3D volume. That's confusing, x and y should not be used for both. Moreover, I believe that PSF_A,B(x,y,z_k) are each 2 dimensional objects. So in reality, spelled out with all the indices, using ^superscript for volume coordinates and _subscript for image coordinates, I believe the equation is

ImgEst_(x',y') = sum {ObjA^(x,y,z_k) conv^(x,y,z_k) PSFA^(x,y,z_k)_(x',y') +.…}

Explaining this equation better, e.g. by writing it out as above or stating that Img_est is a 2D object in image coordinates, and PSF_A,B(x,y,z_k) is 2D in image space contingent on x,y,z_k in volume space, will make this section more understandable.

4) Can the lateral resolution be measured instead of estimated?

5) The manuscript says the reconstruction algorithm is based on optical wave theory. What do the authors mean by this? The algorithm is based on the assumption of a spatially invariant PSF and observations of how to apply Richardon-Lucy to sets of microlenses of different focal lengths. Where does this rely on optical wave theory instead of just classical optics?

6) I assume CAD models of the microlens holder and the autofocus system exists, can these files be made available?

7) Most of the code is said to be available (e.g. real-time behavioral analysis and 3D reconstruction), but in some cases it is not mentioned. Can the code for the tracking and autofocus system be made available?

*Reviewer #3:*

My overall opinion of the manuscript is positive. I think being able to image neuronal activity in a freely moving larval zebrafish is an advance and the current paper serves as a satisfactory proof of principle.

I have some issues regarding the term "whole-brain" and the resolution claimed by the authors. The authors claim, or at least imply, that they can simultaneously (within 1 Orca camera frame which has a 2048 x 2048 pixel sensor) image 800 x 800 x 200 microns at 3.4 x 3.4 x 5 micron resolution. I find this very difficult to believe. Imaging with this resolution requires imaging 800/(3.4/2) = 470 pixels in both x and y for 200/(5/2) = 80 planes (the factor of 2 arise from Nyquist sampling). Given the sensor dimensions one can fit at most 25 planes into it (5 x 5). The authors show that they are able to use their microlens distribution to image 27. I do not believe there is enough information on the chip to have the claimed resolution. The authors may be able to distinguish 2 fluorescent particles 6 microns apart as in Figure 1—figure supplement 7, but these are still sparse particles appearing in the center of their CMOS chip, not a densely fluorescent tissue as a pan-neuronally fluorescent larval zebrafish. I think my argument is corroborated by the data shown in Figure 4: this data does not have the resolution claimed and does not show the whole brain of the larva.

The above argument assumes that the fish's head is also perfectly in focus. The z extent of a larval zebrafish head at this age is ~ 250 microns, which will already be larger than the z field of view. The axial shift shown in Figure 3, typically 20 microns but up to 80 will greatly affect this. The authors mention they use a 500 fps camera for the lateral tracking, but do not (or I missed) the speed of their auto-focus camera for axial tracking: how fast is this?

I do not think that these are "deal-breakers", but I think it is important for the authors to rewrite their claims and be explicit about what their system can't and can do (which is a lot). In the Discussion the authors claim to have developed a whole brain imaging setup: I am not sure what this means.

Figure 1 looks at an agarose restrained larval zebrafish. The authors should be explicit about this in the text and the Figure caption (for example the name of the figure). I do not think that panel d presents maximum intensity projections – they are far too clean for this (the bottom panel looks more like a snapshot of a 3d rendering of the stack). Can the authors correct this in the caption or be explicit about what they are showing?

Closed-loop systems have also been implemented in restrained fish with their tail free to move (e.g. Portugues and Engert 2011) which can remove the issue the authors mention relating to proprioception (Introduction paragraph two). The authors also mention improper vestibular feedback when fish are restrained, but in their setup, due to the closed look, the fish would also experience a reduced vestibular feedback: if the closed-loop was perfect the head would not move at all and the same vestibular deficits would be observed. If this is correct then the authors should comment on this.

The authors talk about "visual stimulation. What does this visual stimulation consist of? This should be explained in the Materials and methods clearly.

The claim in Results paragraph seven is a strong one and I am not sure it is fully warranted. Given the resolution and the data shown I would omit the phrase "for the first time" and again, explain carefully what is meant (here and in other places) by the term "whole brain".

[Editors' note: further revisions were requested prior to acceptance, as described below.]

Thank you for resubmitting your work entitled "Rapid whole brain imaging of neural activity in freely behaving larval zebrafish (*Danio rerio*)" for further consideration at *eLife*. Your revised article has been favorably evaluated by Eve Marder (Senior editor), a Reviewing editor, and two reviewers.

The manuscript has been improved but there are some remaining issues that need to be addressed before acceptance, as outlined below:

This is an exciting manuscript, which reports a new development in Light Field Microscopy (LFM). The authors have made a strong effort at revising the manuscript in response to the last review. They have answered forthrightly almost every point raised and the manuscript is much stronger. There is still one major concern that must be addressed, and there some more minor concerns. The reviews are reproduced in their entirety to aid revision.

Major Concern

1) As brought up by reviewer #3 the definition and technical details of the claimed resolution are not adequately documented and explained. The detailed comments of the reviewer should be fully addressed.

*Reviewer #2:*

The manuscript has improved and I think that most of our comments have been addressed.

The changes include a new piece of useful information: the non-idealness of the point spread function (PSF) has been measured, and attributed, in part, to small differences in focal lengths of the microlenses.

I think this is great work and the revised version is even better. There are a few final comments I'd like to make,

In the subsection “Image reconstruction of XLFM”: "Furthermore, our system took practical conditions, such as imaging system and light properties, into consideration." What does this mean; can this be explained better?

About the reconstruction algorithm: The authors opt for sticking with the conventions and use the same indices x,y on both sides of the equation. This is ok, but in that case, some explanation should be added. For example,

"The 2D convolution is over x and y" and "Per convention in optics, x,y on both sides represent object space, even though in practice, x,y at the left will refer to image space on the camera chip and x,y on the right to the sample coordinates."

In the same section the authors seem to agree that the statement that the algorithm can deal with overlapping fish images depends on the invariance of the PSF, this information should be explicitly stated, i.e. include a statement like "under the condition that the PSF is spatially invariant, which is satisfied apart from small aberrations, the algorithm can handle overlapping fish images".

Results paragraph one: "Therefore, a spatially invariant point spread function (PSF) of the entire optical imaging system could be defined and measured (Figure 1—figure supplement 2)". Here also it would be good to mention that it's an approximately spatially invariant PSF.

A reference to a recept paper from the Vaziri lab, Nobauer et al., 2017, should be included. (This is also about light field microscopy, but I want to emphasize that this does in no way diminish the impact of the current manuscript.)

*Reviewer #3:*

As I mentioned previously, I like the paper, and the authors have addressed most of the minor issues appropriately. There are two points which I am still not sure have been resolved.

1) I do not believe the authors have fully addressed my previous point relating to resolution, which I reproduce below. This argument may be wrong, and I would be very happy if the authors could explain to me where my logic fails.

"I have some issues regarding the term "whole-brain" and the resolution claimed by the authors. The authors claim, or at least imply, that they can simultaneously (within 1 Orca camera frame which has a 2048 x 2048 pixel sensor) image 800 x 800 x 200 microns at 3.4 x 3.4 x 5 micron resolution. I find this very difficult to believe. Imaging with this resolution requires imaging 800/(3.4/2) = 470 pixels in both x and y for 200/(5/2) = 80 planes (the factor of 2 arise from Nyquist sampling). Given the sensor dimensions one can fit at most 25 planes into it (5 x 5). The authors show that they are able to use their microlens distribution to image 27.”

I am not worried about reconstructing the volume; that is not the point here. The issue is that of resolution and discriminability of points. This involves two aspects: the "optical resolution" of the imaging system and the sampling. The Rayleigh resolution criterion states the minimal distance resolvable is half the width of the first diffraction order. This depends on the wavelength of the light and the NA of the system. Ideal sampling is then obtained by using the Nyquist criterion: the inverse image of a pixel should be half the optical resolution of the system, so that there is a "negative dip" in between the two bright pixels.

Using this last sampling definition, the number of resolvable points can be estimated as follows:

- every bright pixel has to be surrounded (in the chip) by a dark set of pixels.

- the number of bright pixels you can have on the chip in this configuration is the number of resolvable points. This is a quarter of the number of pixels on the chip.

- ideally, the inverse image of a pixel should be a quarter of the first diffraction order peak width, for sampling and optical resolution to be perfectly matched.

With a 2048 by 2048 chip, there are at most ~ 1 million resolvable points.

The authors claim they can resolve ~ 2.2 million (800 x 800 x 200 microns at 3.4 x 3.4 x 5 micron resolution).

Deconvolution is a linear process, so XLFM may "shuffle or combine" the intensities of different pixels, but must do so in a linear way.

In addition, as the authors state, the NA of the objective (nominally 1.05) is greatly decreased (to an effective 0.4) because a lot of light is blocked by the array casing. The optical resolution of the system is bound to be significantly affected.

Figure 1—figure supplement 7 does try to address this issue, but I remain unconvinced because column 3 of panel b shows no dip whatsoever in between the two particles. This indicates to me that the authors are not using the Rayleigh criterion. It is definitely possible to separate two identical Gaussians whose positions differ by less than two SDs (situation which approximates to the Rayleigh criterion), but this is not what is usually called resolution.

2) I am not sure I understand the authors' claims of a spatially invariant PSF. In Figure 1—figure supplement 9 they in fact show it is not spatially invariant even within the focal plane (which they attribute to variation in the magnification across microlenses).

[Editors' note: further revisions were requested prior to acceptance, as described below.]

Thank you for resubmitting your work entitled "Rapid whole brain imaging of neural activity in freely behaving larval zebrafish (*Danio rerio*)" for further consideration at *eLife*. Your revised article has been favorably evaluated by Eve Marder (Senior editor), a Reviewing editor, and one reviewer.

The manuscript has been improved but there are some remaining issues that need to be addressed before acceptance, as outlined below. Given that this would be a third round, we must now bring this process to a close with no possibility of further revisions.

Reviewer #3 concerns must be addressed more fully.

I appreciate the authors' comments to the points I raised, but unfortunately I do not feel they have been addressed. I still believe there are two points which need to be addressed: the resolution and the PSF. I think both these points could be addressed by rewriting the manuscript and making claims that are both theoretically sound and supported by the data presented. Given that the manuscript is a proof of concept about an exciting and promising new imaging technique I think it is fundamental to be precise, as this paper will set the baseline for all future work involving this technique and other freely swimming whole brain imaging approaches.

1) Resolution.

The authors have not addressed my argument or provided a counterargument. I still believe my argument is correct. I think the authors are under a misconception. As I mentioned in my previous report, the resolution of an optical system depends on only two things:

a) The optical resolution of the system that results in the Rayleigh criterion.b) The sampling resolution that results in the Nyquist criterion.

Specifically, the resolution does not depend on either the reconstruction (as suggested by the authors in answer to my first review) or the sparseness (as the authors argue in the answer to my second and third review).

I repeat the basis of my argument. Given the dimensions of the chip, the number of resolvable points cannot be more than ~ (2000/2)^2 which is about 1 million. This assumes that a diffraction-limited point in the sample is images onto 1 pixel in the camera chip and that the whole chip is used. The first point is not true for the current system (see Figure 1—figure supplement 9 in which a 0.5um bead results in at least 6 reconstructed pixels in the image). In fact, a one such diffraction limited spot will automatically be imaged onto 27 points just by construction). The second point is also not true for this system (see Figure 1—figure supplement 5 in which only about half the chip is used). My estimate is that at most one fifth of this upper limit of the number of points can be resolved, about 200,000, and probably much closer to 100,000.

Paragraph two of the Results is now very confusing. In the second sentence, the authors state that they can image a field of view of ~ 800 x 800 x 200 μm with 3.4 x 3.4 x 5 μm resolution. This corresponds to 2 million resolvable points. This is probably a factor of 20 out. Then later in the paragraph it is claimed that it is 500 x 500 x 100 μm at this resolution. This corresponds to 400,000 points. First of all I think this is still way too high. And secondly I do not understand the contradictory statements in this paragraph. The resolution does not depend on the sparseness. In addition, one can interpolate and reconstruct at whatever desired resolution to make pretty images, so this does not play a role either.

2) PSF.

The authors have developed a deconvolution algorithm that to my understanding calculates an effective PSF which they assume to be spatially invariant (in 3 dimensions) and which they then use to deconvolve their data. If this is true I think this is a statement that I am very happy with and support. In traditional lightfield microscopy it can be shown theoretically that the PSF is not spatially invariant. I cannot see why the setup presented by the authors would result in a spatially invariant PSF. If they claim this it should be shown theoretically (this is a methods paper). I agree that inhomogeneities arising from unequal microlens magnifications will contribute to worsen the spatial invariance of the PSF, as argued in Figure 1—figure supplement 8. But I do not believe this is the only source for them, so I do not agree with the statement in the figure legend. In Figure 1—figure supplement 9 the authors show that the PSF is not spatially invariant. And the data here relates to the "focal plane z=0" How does the PSF look at x=400um and z =100um? It will most likely look "worse" than that in the last panel of this figure. The authors seem to propose that the PSF of their imaging system is spatially invariant until it is not, away from the center of the field of view and the focal plane. This is not a rigorous scientific statement and not one which can be made in a methods paper that proposes a new imaging technique in a highly regarded journal such as *eLife*. I can definitely stand and support the argument put forward in the first sentence of this paragraph but if the authors claim spatial invariance they will have to either theoretically prove it or measure the PSF throughout the field of view and show it (and Figure 1—figure supplement 9 contradicts this).

---

## [Author Response]

Reviewer #1:1) The chamber in which the freely moving larva swims is ONLY 0.8 mm deep. Thus the animal is sandwiched between glass plates with no real ability to move in the z-direction. Essentially, it moves in two dimensions. The authors should address this limitation in their approach.

We agree with the reviewer that leaving more space in the z direction would be beneficial. Nevertheless, another important factor we must consider is the tracking speed. The traveling range and the moving speed in z-direction are two parameters that *cannot* be easily optimized simultaneously in commercially available products. Here, we chose to use a piezo scanner (Physik Instrumente, P-725KHDS) with a good combination of traveling range (400 μm) and moving speed (330 Hz resonance frequency in the absence of load). In our experimental setup, larva zebrafish, which is typically ~400 μm thick, was swimming in an 800 μm deep chamber, and it had 400 μm free space to explore along the z direction, which can be covered by the 400 μm traveling range moving stage. The prey capture behavior in the larval zebrafish appeared to be normal in such a semi-2D environment.

Tracking in the z direction can be improved in the future. A traveling range beyond 1 mm with sufficiently fast dynamics along the z direction requires a new motion control system. So far, we haven’t explored in this direction. We have added related discussion in the manuscript and addressed the limitation of our approach (Discussion, fifth paragraph).

Reviewer #2:In Cong et al., two advances are reported. First, a tracking system is introduced capable of keeping a freely swimming larval zebrafish in one location most of the time. Second, a new form of light field microscopy is reported capable of fast 3D imaging. Putting these together constitutes a system for whole-brain imaging in freely swimming zebrafish larvae, with a resolution slightly below single-cell.In my opinion this is a major advance and I am supportive of publication in eLife with a few improvements.For background, previous efforts to perform whole-brain imaging in behaving animals consisted of light-sheet (slower than light-field) in head-restrained animals, or light-field (a variant with, I believe, lower spatial resolution) in head-restrained animals. Imaging in freely behaving animals has been done in C. elegans, which move more slowly than zebrafish. Thus, compared to previous work, the advances of this manuscript are considerable. Furthermore, imaging most of the brain in freely swimming animals in really impressive.Points that should be addressed:1) The authors claim that the point spread function (PSF) is spatially invariant. This appears to be true if one considers the microscope an ideal optical system, but with non-ideal optics, it's unlikely. Even with a good objective, the entire system contributes to the PSF, and it's unlikely that all microlenses are diffraction limited over the entire field of view. Moreover, differential distortion between the sub-images would cause the full camera PSF to warp as the point source moves in the sample. So if Richardson-Lucy deconvolution only works with a spatially invariant PSF, and the true PSF is not fully spatially invariant, the question arises, What image artifacts do you get?There may be multiple ways to answer this question. One path might be to (a) move a bead around a few x-y locations, including the extreme ones, and check how spatially invariant the PSF really is (and include the raw PSF volumes in the manuscript, e.g. by measuring them at two extreme points, shifting one by the predicted amount, and overlaying the two in different colors). (b) Next, assuming a spatially invariant PSF derived from one of the bead locations (e.g. the center), reconstruct a bead positioned at various points, including the edges of the volume, and quantify the spread of the point source in the reconstructed volume (this should have high brightness at the original bead position, plus dimmer pixel values spread at other locations, which should be quantified).

We agree with the reviewer that only an ideal imaging system would have a spatially invariant PSF and it is informative to characterize the imaging performance under realistic conditions. We performed calibration in the way the reviewer suggested and confirmed that the spatial invariance could not be perfectly conserved across the entire imaging volume. We further confirmed that the spatial variance of PSF was mainly due to the focal length variation of the customized micro-lenses (Figure 1—figure supplement 8). We have clarified this point in the Results, second paragraph.

This problem can be solved either by employing more precisely machined micro-lenses or a generalized reconstruction algorithm. The generalized reconstruction will take 27 PSFs measured from 27 micro-lenses, instead of 2 PSFs measured from two groups of micro-lenses, as in our current implementation. For accurate reconstruction, the magnification factor for each micro-lens should be characterized experimentally to account for the focal length variation. In this way, the reconstruction will be more accurate. However, the increased computational complexity cannot be handled by our current computing platform. Further optimization of XLFM will be under future investigation.

We also reconstructed beads that were placed at extreme points in the imaging volume using a PSF that was measured by placing a fluorescent particle at the center of the field of view. We found that reconstructions resulted in nicely localized spots within a field of view of 500 μm in diameter but were distorted near the edge of the imaging volume. This is apparently due to the spatial variance of the PSF, as suggested by the reviewer.

In summary, we appreciate the reviewer’s comments, which help us identify the focal length variation of the customized micro-lenses as the major contribution to the spatial variance of PSF. We envisioned two ways to solve this problem: (1) a more precisely machined microlenses array; (2) employing generalized reconstruction algorithm in which each micro-lens is characterized individually. Both directions will be investigated in the future.

2) A follow-up: if the PSF is not fully spatially invariant, what does this mean for the statement that overlapping sub-images are permitted (subsection “Image reconstruction of XLFM”)? My understanding is that the overlap is fine so long as the PSF is fully x-y invariant, and if not, then some artifacts will be introduced. The reasons and assumptions underlying this statement should be clarified in the text.

We agree that spatial invariance is important for correct reconstruction over the overlapping regions. However, our current implementation cannot produce a perfectly spatially invariant PSF. If the sample is not sparse, spatially variant PSF could lead to reconstruction artifacts, which is not easy to quantify. The best way to solve this problem would be to build the next generation XLFM and to make a direct comparison with the current one.

For clarity, points (1) and (2) are not criticisms of the system, only a call for characterization of the artifacts that the reconstruction algorithm introduces when using simplifying assumptions.3) The reconstruction algorithm (subsection “Image reconstruction of XLFM”) contains confusing notation (if I understand it correctly). The coordinates (x,y) on the left hand side of Equation 5, refer to image coordinates. But (x,y,z_k) on the right hand side refers to coordinates in the 3D volume. That's confusing, x and y should not be used for both. Moreover, I believe that PSF_A,B(x,y,z_k) are each 2 dimensional objects. So in reality, spelled out with all the indices, using ^superscript for volume coordinates and _subscript for image coordinates, I believe the equation isImgEst_(x',y') = sum {ObjA^(x,y,z_k) conv^(x,y,z_k) PSFA^(x,y,z_k)_(x',y') +.…}Explaining this equation better, e.g. by writing it out as above or stating that Img_est is a 2D object in image coordinates, and PSF_A,B(x,y,z_k) is 2D in image space contingent on x,y,z_k in volume space, will make this section more understandable.

Thanks for the comment. We believe that the reviewer has fully understood the equation. The notations may sound confusing, but we would like to follow the convention in the field of optical imaging: PSF is conventionally defined in object space even though it is actually measured in imaging space, so that the size of the PSF reflects the imaging resolution. The image captured on the camera is also conventionally transformed and interpreted as in object space because the actual pixel size and the factor of magnification are not important in above equations. In this way, the imaging formation can be conveniently written as the convolution between object and PSF in object space.

4) Can the lateral resolution be measured instead of estimated?

We have added experimental characterization (Figure 1—figure supplement 6).

5) The manuscript says the reconstruction algorithm is based on optical wave theory. What do the authors mean by this? The algorithm is based on the assumption of a spatially invariant PSF and observations of how to apply Richardon-Lucy to sets of microlenses of different focal lengths. Where does this rely on optical wave theory instead of just classical optics?

The optical wave theory we referred here is to distinguish it from classical optics, which is often called light ray optics. In conventional light field microscopy (LFM), the reconstruction is based on the light ray assumption, which is implied in the name of “light field”. However, the light ray assumption cannot account for the limitations of resolution and the depth of view in LFM. To take these effects into account accurately, optical diffraction described by the optical wave theory needs to be incorporated into the reconstruction algorithm. Actually, M. Broxton et al. had introduced a way of doing so in the conventional LFM, as seen in Broxton et al., 2013. In XLFM, we defined a spatially invariant PSF (assuming an ideal imaging system), which reflects the resolution limit and the beam diffraction effect. For this reason, we claimed that the XLFM reconstruction algorithm was based on optical wave theory to distinguish it from conventional LFM reconstruction algorithm.

6) I assume CAD models of the microlens holder and the autofocus system exists, can these files be made available?

We have added CAD models of the microlens holder for fluorescence imaging and autofocus.

7) Most of the code is said to be available (e.g. real-time behavioral analysis and 3D reconstruction), but in some cases it is not mentioned. Can the code for the tracking and autofocus system be made available?

The code is available in the supplementary software.

Reviewer #3:My overall opinion of the manuscript is positive. I think being able to image neuronal activity in a freely moving larval zebrafish is an advance and the current paper serves as a satisfactory proof of principle.I have some issues regarding the term "whole-brain" and the resolution claimed by the authors. The authors claim, or at least imply, that they can simultaneously (within 1 Orca camera frame which has a 2048 x 2048 pixel sensor) image 800 x 800 x 200 microns at 3.4 x 3.4 x 5 micron resolution. I find this very difficult to believe. Imaging with this resolution requires imaging 800/(3.4/2) = 470 pixels in both x and y for 200/(5/2) = 80 planes (the factor of 2 arise from Nyquist sampling). Given the sensor dimensions one can fit at most 25 planes into it (5 x 5). The authors show that they are able to use their microlens distribution to image 27. I do not believe there is enough information on the chip to have the claimed resolution. The authors may be able to distinguish 2 fluorescent particles 6 microns apart as in Figure 1—figure supplement 7, but these are still sparse particles appearing in the center of their CMOS chip, not a densely fluorescent tissue as a pan-neuronally fluorescent larval zebrafish. I think my argument is corroborated by the data shown in Figure 4: this data does not have the resolution claimed and does not show the whole brain of the larva.The above argument assumes that the fish's head is also perfectly in focus. The z extent of a larval zebrafish head at this age is ~ 250 microns, which will already be larger than the z field of view. The axial shift shown in Figure 3, typically 20 microns but up to 80 will greatly affect this. The authors mention they use a 500 fps camera for the lateral tracking, but do not (or I missed) the speed of their auto-focus camera for axial tracking: how fast is this?

1) XLFM can cover a volume larger than 200 μm in the z direction. All data shown in the manuscript were reconstructed over a volume of 800 μm x 800 μm x 400 μm and were cropped later to remove empty space for better display. As shown in the reconstruction algorithm, there was no constraint on the number of z planes to be reconstructed. Since we measured PSF over 200 planes with 2 μm interspacing, the reconstruction was done over the same z range of 400 μm. Therefore, the whole brain of the larval zebrafish was indeed covered by XLFM. We have clarified this point in the Results, second paragraph.

2) We appreciate the reviewer’s comment on the imaging resolution. Indeed, the sparseness of neuronal activities (or sparsely labeled neurons) is a prerequisite for obtaining both high resolution and large field of view. The relationship between resolution and sparseness of neural activity was discussed and added to the manuscript and summarized in Figure 1—figure supplement 11. In short, we introduced a sparseness index ρ, defined as the fraction of neurons that areactivated at a given instant. Given the total number of neurons (~ 80,000) in the larval zebrafish brain, we performed computer simulation and identified a critical ρ = 0.11, below which neuronal activities can be resolved at optimal resolution (Figure 1—figure supplement 11). When population activity is denser, XLFM would obtain a more coarse-grained neural activity map with reduced resolution (Figure 1—figure supplement 11). We have clarified this point in the Results, third paragraph, and in Discussion, third paragraph.

3) Thanks for pointing out the error. The correct acquisition parameters for the lateral tracking camera should be 2 ms exposure time and 300 Hz (or higher) frame rate, which is consistent with the claimed lateral tracking update rate of 300 Hz. The axial tracking camera ran at 10 ms exposure time and 100 Hz frame rate, which is also consistent with claimed axial tracking update rate of 100 Hz. We have corrected and included more information in the manuscript, see Materials and methods.

I do not think that these are "deal-breakers", but I think it is important for the authors to rewrite their claims and be explicit about what their system can't and can do (which is a lot). In the Discussion the authors claim to have developed a whole brain imaging setup: I am not sure what this means.Figure 1 looks at an agarose restrained larval zebrafish. The authors should be explicit about this in the text and the Figure caption (for example the name of the figure). I do not think that panel d presents maximum intensity projections – they are far too clean for this (the bottom panel looks more like a snapshot of a 3d rendering of the stack). Can the authors correct this in the caption or be explicit about what they are showing?

Thanks for the comment. We explicitly mentioned the agarose-restrained condition in the main text as well as in the Figure 1 caption. Because the main theme of Figure 1 is to introduce the principle of XLFM and to demonstrate its capabilities for volume imaging in larval zebrafish, we think it might be better to keep the figure title unchanged.

Panel D in Figure 1 shows maximum intensity projections of time series 3D volume images. In other words, we performed maximum intensity projections on space (top, top view; bottom, side view) and time. We have clarified this point in the figure caption accordingly.

Closed-loop systems have also been implemented in restrained fish with their tail free to move (e.g. Portugues and Engert 2011) which can remove the issue the authors mention relating to proprioception (Introduction paragraph two). The authors also mention improper vestibular feedback when fish are restrained, but in their setup, due to the closed look, the fish would also experience a reduced vestibular feedback: if the closed-loop was perfect the head would not move at all and the same vestibular deficits would be observed. If this is correct then the authors should comment on this.

We agree with the reviewer that the head-restrained and tail-free setting is a simple and elegant behavioral paradigm for incorporating multiple sensory cues, such as proprioception, and for studying sensorimotor transformation in larval zebrafish. We have included references to related works and added discussion on these in the manuscript (see Discussion, second paragraph).

We also agree that in our closed-loop setup, any interpretation of behaviors and neural activity associated with self-motion must take into account motion compensation driven by the tracking system. Indeed, the perception of linear acceleration, encoded by the vestibular feedback, would be significantly reduced. The perceptions of angular acceleration and the relative velocity of water flow may remain intact. We have added one paragraph discussing the limitation of our approach and future improvement of the tracking system. See Discussion, fourth paragraph.

The authors talk about "visual stimulation. What does this visual stimulation consist of? This should be explained in the Materials and methods clearly.

Thanks! Detailed information about visual stimulation has been added in the Materials and methods section.

The claim in Results paragraph seven is a strong one and I am not sure it is fully warranted. Given the resolution and the data shown I would omit the phrase "for the first time" and again, explain carefully what is meant (here and in other places) by the term "whole brain".

Thanks! We have deleted this phrase. We agree that our writing may be misleading. “Whole brain” means that our imaging volume reconstruction can cover the entire larval zebrafish head. However, to achieve close to single neuron resolution, the population neural activity must be sparse. We have clarified our essential claims in the introduction of XLFM.

[Editors' note: further revisions were requested prior to acceptance, as described below.]

Reviewer #2:The manuscript has improved and I think that most of our comments have been addressed.The changes include a new piece of useful information: the non-idealness of the point spread function (PSF) has been measured, and attributed, in part, to small differences in focal lengths of the microlenses.I think this is great work and the revised version is even better. There are a few final comments I'd like to make,In the subsection “Image reconstruction of XLFM”, "Furthermore, our system took practical conditions, such as imaging system and light properties, into consideration." What does this mean; can this be explained better?

Thanks for comment. As compared with a theoretically derived PSF, the experimentally measured one account for practically conditions, such as a non-ideal imaging objective, actual positions of individual micro-lenses, the actual spectrum of received fluorescence signal et al. We have added more description in the manuscript.

About the reconstruction algorithm: The authors opt for sticking with the conventions and use the same indices x,y on both sides of the equation. This is ok, but in that case, some explanation should be added. For example,"The 2D convolution is over x and y" and "Per convention in optics, x,y on both sides represent object space, even though in practice, x,y at the left will refer to image space on the camera chip and x,y on the right to the sample coordinates."

Thanks for the comment. We have modified the manuscript accordingly.

In the same section the authors seem to agree that the statement that the algorithm can deal with overlapping fish images depends on the invariance of the PSF, this information should be explicitly stated, i.e. include a statement like "under the condition that the PSF is spatially invariant, which is satisfied apart from small aberrations, the algorithm can handle overlapping fish images".

Thanks for the comment. We have modified the manuscript accordingly.

Results paragraph one: "Therefore, a spatially invariant point spread function (PSF) of the entire optical imaging system could be defined and measured (Figure 1—figure supplement 2)". Here also it would be good to mention that it's an approximately spatially invariant PSF.

Thanks for the comment. We have modified the manuscript accordingly.

A reference to a recent paper from the Vaziri lab, Nobauer et al., 2017, should be included. (This is also about light field microscopy, but I want to emphasize that this does in no way diminish the impact of the current manuscript.)

Thanks for the comment. We have updated the manuscript.

Reviewer #3:As I mentioned previously, I like the paper, and the authors have addressed most of the minor issues appropriately. There are two points which I am still not sure have been resolved.1) I do not believe the authors have fully addressed my previous point relating to resolution, which I reproduce below. This argument may be wrong, and I would be very happy if the authors could explain to me where my logic fails."I have some issues regarding the term "whole-brain" and the resolution claimed by the authors. The authors claim, or at least imply, that they can simultaneously (within 1 Orca camera frame which has a 2048 x 2048 pixel sensor) image 800 x 800 x 200 microns at 3.4 x 3.4 x 5 micron resolution. I find this very difficult to believe. Imaging with this resolution requires imaging 800/(3.4/2) = 470 pixels in both x and y for 200/(5/2) = 80 planes (the factor of 2 arise from Nyquist sampling). Given the sensor dimensions one can fit at most 25 planes into it (5 x 5). The authors show that they are able to use their microlens distribution to image 27."I am not worried about reconstructing the volume; that is not the point here. The issue is that of resolution and discriminability of points. This involves two aspects: the "optical resolution" of the imaging system and the sampling. The Rayleigh resolution criterion states the minimal distance resolvable is half the width of the first diffraction order. This depends on the wavelength of the light and the NA of the system. Ideal sampling is then obtained by using the Nyquist criterion: the inverse image of a pixel should be half the optical resolution of the system, so that there is a "negative dip" in between the two bright pixels.Using this last sampling definition, the number of resolvable points can be estimated as follows:- every bright pixel has to be surrounded (in the chip) by a dark set of pixels.- the number of bright pixels you can have on the chip in this configuration is the number of resolvable points. This is a quarter of the number of pixels on the chip.- ideally, the inverse image of a pixel should be a quarter of the first diffraction order peak width, for sampling and optical resolution to be perfectly matched.With a 2048 by 2048 chip, there are at most ~ 1 million resolvable points.The authors claim they can resolve ~ 2.2 million (800 x 800 x 200 microns at 3.4 x 3.4 x 5 micron resolution).Deconvolution is a linear process, so XLFM may "shuffle or combine" the intensities of different pixels, but must do so in a linear way.In addition, as the authors state, the NA of the objective (nominally 1.05) is greatly decreased (to an effective 0.4) because a lot of light is blocked by the array casing. The optical resolution of the system is bound to be significantly affected.Figure 1—figure supplement 7 does try to address this issue, but I remain unconvinced because column 3 of panel b shows no dip whatsoever in between the two particles. This indicates to me that the authors are not using the Rayleigh criterion. It is definitely possible to separate two identical Gaussians whose positions differ by less than two SDs (situation which approximates to the Rayleigh criterion), but this is not what is usually called resolution.

Thanks for the comment.

We agree with reviewer’s comment that that a total number of resolvable points is limited by the total number of pixels on image sensor. We thus responded that the sparsity was a prerequisite for obtaining both high resolution and large field of view. By doing simulation, we found that when the sparseness index ρ, defined as the fraction of total neuron population (~80,000 in larval zebrafish), was less than 0.11, corresponding to 8,800 neurons distributed over a larval zebrafish brain, individual neurons could be resolved with an optimal resolution of 3.4 μm x 3.4 μm x 5 μm, as shown in Figure 1—figure supplement 11.

In the extreme case when there were only two particles separated in z direction, as shown in Figure 1—figure supplement 7, Figure 1—figure supplement 27 sub-images of the same two particles were captured, which provided partially redundant information for these two particles. As a result, these two particles could be resolved at optimal resolution. The theoretical analysis of this optimal resolution was provided in the Materials and methods section of “Resolution characterization of XLFM”. It was also experimentally confirmed, as shown in Figure 1—figure supplement 6 and Figure 1—figure supplement 7.

To respond to reviewer’s concern on Figure 1—figure supplement 7, we characterized resolution by spatial frequency analysis (Figure 1—figure supplement 7 column 4), which is an important and precise way to characterize the resolution. But we agree that it would be more convincing to see a dip between two particles. Because higher spatial frequency component has much lower signal to noise ratio than lower spatial frequency component, as shown in Figure 1—figure supplement 7 column 4, it may not be easy to directly see a dip in raw images. Therefore, raw images are conventionally deconvolved to assist visualization. In the updated Figure 1—figure supplement 7, we added an extra column to show the results after deconvolution using linear Wiener filtering method. The expected dips between two particles were much more evident in the deconvolved images than that before deconvolution.

Our reconstruction method, which was developed from Richardson-Lucy deconvolution, inherited the same property that there was no limit on the number of voxels to be reconstructed: all voxels were estimated with maximum likelihood. However, as the reviewer correctly pointed out, there was no information gain during the reconstruction. As a result, although we could reconstruct 2.2 million voxels of 1.7 μm × 1.7 μm × 2 μm over the imaging volume of Ф 800 μm × 200 μm, these voxels were not completely independent variables. However, when sample was sparse, the small fraction of nonzero voxels can be treated independently, and by using Nyquist sampling and keeping the voxel small, we could achieve the optimal resolution at 3.4 μm x 3.4 μm x 5 μm. An extreme case was when there were only two particles separated in z direction, as shown in Figure 1—figure supplement 7 and discussed above.

When sample was dense, nearby voxels would not be independent anymore because the captured information was insufficient to assign independent value for each voxel. This resulted in degraded resolution, as shown in Figure 1—figure supplement 11. Together, given the limited number of pixels on the image sensor implemented in our setup, the optimal resolution could only be achieved when the sample was sparse. We apologize for possible misleading statements. We have clarified this point in the main text.

Throughout the manuscript, we used Abbe limitd=0.5λNA for resolution characterization, which differs slightly from Rayleigh criteriond=0.61λNA in an optical system with circular apertures.

The effective NA of 0.4 we mentioned in the manuscript is defined based on light collection efficiency. It means the light collection efficiency of this system is equivalent to the one using an objective of 0.4 NA. This collection efficiency could be improved by using more micro-lenses in the array, but it also requires more camera pixels to ensure certain field of view for each micro-lens. This collection efficiency argument is not applicable for resolution comparison.

2) I am not sure I understand the authors' claims of a spatially invariant PSF. In Figure 1—figure supplement 9 they in fact show it is not spatially invariant even within the focal plane (which they attribute to variation in the magnification across microlenses).

Our statement actually is that a spatially invariant PSF can be defined if the optical system is ideal. Our current implementation, however, is not perfect as shown in Figure 1—figure supplement 9. We have modified the text to clarify this point.

[Editors' note: further revisions were requested prior to acceptance, as described below.]

Reviewer #3 concerns must be addressed more fully.I appreciate the authors' comments to the points I raised, but unfortunately I do not feel they have been addressed. I still believe there are two points which need to be addressed: the resolution and the PSF. I think both these points could be addressed by rewriting the manuscript and making claims that are both theoretically sound and supported by the data presented. Given that the manuscript is a proof of concept about an exciting and promising new imaging technique I think it is fundamental to be precise, as this paper will set the baseline for all future work involving this technique and other freely swimming whole brain imaging approaches.1) Resolution.The authors have not addressed my argument or provided a counterargument. I still believe my argument is correct. I think the authors are under a misconception. As I mentioned in my previous report, the resolution of an optical system depends on only two things:a) The optical resolution of the system that results in the Rayleigh criterion.b) The sampling resolution that results in the Nyquist criterion.Specifically, the resolution does not depend on either the reconstruction (as suggested by the authors in answer to my first review) or the sparseness (as the authors argue in the answer to my second and third review).I repeat the basis of my argument. Given the dimensions of the chip, the number of resolvable points cannot be more than ~ (2000/2)^2 which is about 1 million. This assumes that a diffraction-limited point in the sample is images onto 1 pixel in the camera chip and that the whole chip is used. The first point is not true for the current system (see Figure 1—figure supplement 9 in which a 0.5um bead results in at least 6 reconstructed pixels in the image). In fact, a one such diffraction limited spot will automatically be imaged onto 27 points just by construction). The second point is also not true for this system (see Figure 1—figure supplement 5 in which only about half the chip is used). My estimate is that at most one fifth of this upper limit of the number of points can be resolved, about 200,000, and probably much closer to 100,000.Paragraph two of the Results is now very confusing. In the second sentence, the authors state that they can image a field of view of ~ 800 x 800 x 200 μm with 3.4 x 3.4 x 5 μm resolution. This corresponds to 2 million resolvable points. This is probably a factor of 20 out. Then later in the paragraph it is claimed that it is 500 x 500 x 100 μm at this resolution. This corresponds to 400,000 points. First of all I think this is still way too high. And secondly I do not understand the contradictory statements in this paragraph. The resolution does not depend on the sparseness. In addition, one can interpolate and reconstruct at whatever desired resolution to make pretty images, so this does not play a role either.

Thanks for the comment. We would like to try our best to clarify our statement and avoid possible misunderstanding.

We believe that the major disagreement between us might be due to our different understanding on the claim, that is, XLFM could achieve optimal resolution of 3.4 x 3.4 x 5 μm within the volume of Ø 800 x 200 μm when the sample is sparse.

The reviewer argued that resolution is independent of sparsity and our claim thus meant that all *N* ≈ 2 million voxels in this volume should be measured independently, as imposed by Nyquist sampling theorem. On the other hand, we argued that sparsity constraint was central to this claim. If there is no sparsity constraint, we would agree that the Nyquist sampling is required. However, if sparsity constraint is imposed, which means the number of non-zero voxels is far less than the total number of voxels in the imaging volume, our reconstruction algorithm can achieve the claimed resolution of 3.4 x 3.4 x 5 μm within the volume of Ø 800 x 200 um. This claim can be illustrated by the following simplified example:

The designed XLFM has 27 micro-lenses. These micro-lenses image the same object from different view angles and form 27 sub-images on a single image sensor chip. Here, we simplify the imaging formation model and assume that each sub-image is captured by an individual camera (Author response image 1). Please note that this model is not a precise description of the actual XLFM system; but it draws a close analogy with XLFM and serves as a valid model to illustrate the design principle of XLFM. Additionally, the example is illustrated in two dimensions, but can be easily generalized to three dimensions.

As shown in Author response image 1, Camera A captures a top view of three pairs of dots within a gray square area. The imaging system of camera A is designed to have relatively high resolution in x direction, but very poor resolution in z direction. The poor resolution in z direction actually means very large depth of view in this direction. As a result, camera A can resolve two laterally aligned and closely spaced particles anywhere within the imaging field of view (gray square). This system, however, do not provide any capability to resolve two particles in z direction.

To make the system capable of resolving particles in z direction, we can add camera B to get a side view of the same object over the same field of view. The camera B provides high resolution in z direction, but poor resolution in x direction. If we combine information from both camera A and B, which are complementary to each other, we can see that this system can resolve two closely spaced particles anywhere within the field of view, as shown in Author response image 1.

Therefore, we claim that the imaging system, which consists of two cameras A and B oriented orthogonal to each other, provides high resolution both in x and z directions over the entire field of view when the sample is sparse.

In the above claim, the sparsity constraint is important because this system has problems to resolve denser samples. As shown in Author response image 1, three different sample distributions i, ii, and iii can generate exactly the same observation on camera A and B. In this case, it is not possible to distinguish whether four particles or two particles are present within the field of view.

The above problem can be alleviated by adding more cameras sampling from different perspectives. By adding camera C, as shown in Author response image 1, the images captured by camera C can provide useful information to distinguish all three different cases.

In the above example, each camera is analogous to one micro-lens in XLFM. In our XLFM, we have 27 micro-lenses sampling from 27 different view angles. As a result, the designed XLFM can handle much more dense samples than that illustrated in Author response image 1.

To summarize, our claim of “XLFM could achieve optimal resolution of 3.4 x 3.4 x 5 μm within the volume of Ø 800 x 200 μm when the sample is sparse” is based on following facts:

1) We showed that each micro-lens with NA of 0.075 provided an in-plane resolution of 3.4 um. This resolution was experimentally confirmed to be preserved throughout the imaging volume of Ø 800 x 200 um, as shown in Figure 1—figure supplement 6.

2) We theoretically showed that the micro-lenses on the edge of the objective’s pupil provide resolution in z direction because the PSFs generated by these micro-lenses are tilted, as shown in Figure 1—figure supplement 2 and discussed in subsection “Resolution characterization of XLFM” in manuscript. Combing this theoretical analysis and result (1), we expected that the z resolution of 4um can be achieved throughout the imaging volume of Ø 800 x 200 um. Due to the limited signal to noise ratio in practical conditions, however, we experimentally obtained a resolution of 5 μm in z direction, as shown in Figure 1—figure supplement 7.

3) Based on the results (1) and (2), we concluded that our designed XLFM can resolve two closely spaced particles anywhere in the imaging volume of Ø 800 x 200 μm with optimal resolution of 3.4 x 3.4 x 5 um.

4) We expected that the capability of resolving objects with optimal resolution can be generalized from the simple case of having only two particles within the field of view to more complicated cases. A detailed theoretical analysis of such generalization is out of the scope of this work. Instead, we did computer simulation and found that the sample sparseness can be a proper indicator of when this optimal resolution can be achieved, as shown in Figure 1—figure supplement 11.

5) By combining the above results, we claimed that our designed XLFM, in the absence of micro-lenses’ focal length variation, can achieve a resolution of 3.4 x 3.4 x 5 μm within the volume of Ø 800 x 200 μm when sample is sparse.

6) Due to the focal length variation, the PSF of the optical system is not fully spatially invariant (see below). As a result, the reconstruction performance is degraded comparing to our initial design (Figure 1—figure supplement 9). Thus we claimed that our current implementation of XLFM achieved optimal resolution of 3.4 x 3.4 x 5 μm within a reduced imaging volume of Ø 500 x 100 um.

Now we have streamlined the structure of the main text based on the logical flow from (1)-(6).

As a side note, our designed XLFM is not the only one that makes use of the sparsity constraint. In the field of compressed sensing, it has been demonstrated that image signals can be recovered with fewer number of measurements than that required by the Shannon-Nyquist sampling theorem when sample is sparse. Strictly speaking, our method is not compressed sensing, but our developed XLFM can also recover high resolution information of sparse objects within a large field of view in the way illustrated in Author response image 1.

2) PSF.The authors have developed a deconvolution algorithm that to my understanding calculates an effective PSF which they assume to be spatially invariant (in 3 dimensions) and which they then use to deconvolve their data. If this is true I think this is a statement that I am very happy with and support. In traditional lightfield microscopy it can be shown theoretically that the PSF is not spatially invariant. I cannot see why the setup presented by the authors would result in a spatially invariant PSF. If they claim this it should be shown theoretically (this is a methods paper). I agree that inhomogeneities arising from unequal microlens magnifications will contribute to worsen the spatial invariance of the PSF, as argued in Figure 1—figure supplement 8. But I do not believe this is the only source for them, so I do not agree with the statement in the figure legend. In Figure 1—figure supplement 9 the authors show that the PSF is not spatially invariant. And the data here relates to the "focal plane z=0" How does the PSF look at x=400um and z =100um? It will most likely look "worse" than that in the last panel of this figure. The authors seem to propose that the PSF of their imaging system is spatially invariant until it is not, away from the center of the field of view and the focal plane. This is not a rigorous scientific statement and not one which can be made in a methods paper that proposes a new imaging technique in a highly regarded journal such as eLife. I can definitely stand and support the argument put forward in the first sentence of this paragraph but if the authors claim spatial invariance they will have to either theoretically prove it or measure the PSF throughout the field of view and show it (and Figure 1—figure supplement 9 contradicts this).

Thanks for the comment. We would like to make our best effort to explain the underlying theory of XLFM. We have added a new section discussing the spatial invariance of the PSF in the Materials and methods of the manuscript.

Since the raw image is in 2D, the spatially invariance of PSF is only required in 2D as well. It’s implied in the reconstruction algorithm. To avoid misunderstanding, we have clarified this in the manuscript.

The definition of a spatially invariant PSF fundamentally means that in an ideal optical microscopy system, the resulting image can be described as a convolution between object and PSF (Introduction to Fourier Optics, Goodman):Image=Object⊗PSF

This equation forms the basis of our reconstruction algorithm, as shown in subsection “Image reconstruction of XLFM” of the manuscript. If the PSF is far from spatially invariant, the imaging reconstruction wouldn’t yield any meaningful result.

One of the fundamental differences between XLFM and conventional LFM is the location of the microlens array. In XLFM, the microlens array is placed at the pupil plane and the image sensor is at imaging plane, whereas in conventional LFM, the microlens array is placed at the image plane and the image sensor is at pupil plane. Only in XLFM, it is possible to define and measure a spatially invariant PSF. The reasons are as follows:

1) Spatially invariant PSFs can be defined for individual sub-imaging systems consisting of different micro-lenses.

As shown in Author response image 2, which is a simplified version of Author response image 1 in the manuscript, the object under the imaging objective is firstly imaged onto an intermediate imaging plane by tube lens 1, and then relayed by tube lens 2 and individual micro-lens A1 and A2 onto the camera image sensor. By definition, the imaging process in an ideal imaging system is linear and spatially invariant, so spatially invariant PSFs for sub-imaging systems consisting of micro-lens A1 and A2 can be defined as:ImageA1=ObjectA1⊗PSFA1ImageA2=ObjectA2⊗PSFA2

As discussed in the Materials and methods, the convolution can be performed either in the object space or imaging space. If we perform the convolution in the imaging space, then the coordinates of ObjectA1 and ObjectA2 should be scaled by the magnification factors of their sub-image systems.

**Author response image 2. respfig2:** Point spread function of XLFM.

2) A spatially invariant PSF can be defined for the entire imaging system if the magnifications of all sub-imaging systems are the same.

Because the camera captured image is the summation of all sub-images, the summation of all PSFs formed by individual micro-lenses can be defined as a single PSF if the magnifications of different sub-images are the same. Below is the simple proof:ImageA=ImageA1+ImageA2=ObjectA1⊗PSFA1+ObjectA2⊗PSFA2

If magnifications of the sub-imaging systems A1 and A2 are the same, then ObjectA1=ObjectA2=Object, and the above equation can be rewritten as:ImageA=Object⊗(PSFA1+PSFA2)=Object⊗PSFAWhere PSFA=PSFA1+PSFA2.

The variation of individual micro-lenses’ focal length indicate that 𝑃𝑆𝐹_𝐴_ or 𝑃𝑆𝐹_𝐵_ (see Materials and methods) are not fully spatially invariant, but it does not affect the imaging formation theory of XLFM. The spatial variance, as measured in Figure 1—figure supplement 8, leads to degraded reconstruction performance, as shown in Figure 1—figure supplement 9. This degradation is negligible near the center of the field of view, but becomes more evident at the edge of the field of view. This is because the PSF is measured near the center of the field of view. The reconstruction algorithm will produce 27 estimates of the same object based on 27 sub-images. In the meanwhile, it tries to combine and align these estimates all together in the same coordinates. The position where the PSF is measured determines the origin of this coordinates. If the magnifications of all sub-images are the same, all estimates can be combined coherently to produce an accurate reconstruction. If magnifications of different micro-lenses are different, the reconstruction will yield an image that is clear near the origin of the coordinates but blurred at the edge, as shown in Author response image 3.

**Author response image 3. respfig3:** Resolution degradation caused by magnification variation of micro-lenses in XLFM.